

# Goodness of fit by Neyman-Pearson testing

Gaia Grosso[1,2,3,4,5]⋆, Marco Letizia[6], Maurizio Pierini[2] and Andrea Wulzer[7]

**1** Dipartimento di Fisica e Astronomia, Università di Padova and INFN,
Sezione di Padova, Via Francesco Marzolo, 8, 35121 Padova, Italy
**2** Experimental Physics department, CERN,
Espl. des Particules 1, 1211 Meyrin, Geneva, Switzerland
**3** NSF AI Institute for Artificial Intelligence and Fundamental Interactions,
77 Massachusetts Avenue, 26-505 Cambridge, MA 02139, USA
**4** MIT Laboratory for Nuclear Science, 77 Massachusetts Avenue,
26-505 Cambridge, MA 02139, USA
**5** School of Engineering and Applied Sciences, Harvard University,
150 Western Avenue, Allston, MA 02134, USA
**6** MaLGa Center - DIBRIS, Università di Genova and INFN,
Sezione di Genova, Via Dodecaneso 35, 16146 Genova, Italy
**7** IFAE, BIST, Campus UAB, 08193 Bellaterra, Barcelona, Spain and ICREA,
Passeig de Lluís Companys 23, 08010 Barcelona, Spain

⋆ gaia.grosso@cern.ch

## Abstract

The Neyman–Pearson strategy for hypothesis testing can be employed for goodness of fit if the alternative hypothesis is selected from data by exploring a rich parametrised family of models, while controlling the impact of statistical fluctuations. The *New Physics Learning Machine* (NPLM) methodology has been developed as a concrete implementation of this idea, to target the detection of new physical effects in the context of high energy physics collider experiments. In this paper we conduct a comparison of this approach to goodness of fit with others, in particular with classifier-based strategies that share strong similarities with NPLM. From our comparison, NPLM emerges as the more sensitive test to small departures of the data from the expected distribution and not biased towards detecting specific types of anomalies. These features make it suited for agnostic searches for new physics at collider experiments. Its deployment in other scientific and industrial scenarios should be investigated.



# 1   Introduction

Testing the consistency between a set $\mathcal{D} = \{x_i\}_{i=1}^{N_\mathcal{D}}$ of realizations of a random variable $x$, and one hypothesis of Reference (R) for its expected distribution, is a problem known in statistics as *goodness of fit* (GoF). This task is ubiquitous and GoF methods find application in innumerable areas of science and technology.

A primary task in high energy physics is to assess whether experimental measurements are distributed according to the prediction of the Standard Model (SM) of particle physics. An anomalous data behaviour would signal the existence of new fundamental physical laws—the so-called New Physics—supplementing or replacing the SM laws. In high energy physics as well as in many other domains, GoF emerges as a necessity in order to assess and compare computer codes that provide the theoretical predictions for data distributions. Evaluating the quality of generative models, in the context of computer science, can also be formulated as a GoF problem. The task of monitoring complex apparatuses, such as particle detectors, can be addressed in this framework as well. An overview of the vast GoF literature goes beyond the scope of the present article. See [1, 2] for references and a concise overview from a high energy physics perspective.

**Classifier-based goodness of fit**   A problem that is conceptually different but practically related to GoF is the one of *two-sample testing* (2ST). In 2ST, one is given a second set of data that we denote as $\mathcal{R} = \{x_i\}_{i=1}^{N_\mathcal{R}}$, and aims at assessing whether or not the $\mathcal{D}$ and the $\mathcal{R}$ data sets are drawn from the same (unknown) statistical distribution. The connection with GoF stems from the fact that in many applications of practical relevance the data distribution in the reference hypothesis is not known in closed form. The only available representation of the R hypothesis is provided by a set of instances of the variable $x$ that are known to follow the R distribution, i.e. by a reference data set $\mathcal{R}$. Depending on the specific application, $\mathcal{R}$ can be obtained through simulations, by collecting a set of control measurements, or combinations thereof. The GoF comparison between the data $\mathcal{D}$ and the distribution in the R hypothesis

is carried out in practice as a 2ST between $\mathcal{D}$ and the reference data set $\mathcal{R}$. The GoF test is thus effectively a type of 2ST where the two data sets $\mathcal{D}$ and $\mathcal{R}$ play an asymmetric role. In principle, and in many practical applications, there is no obstruction to increase the size of the $\mathcal{R}$ set—for instance by synthesizing more artificial data—in order to offer a more faithful representation of the R distribution. When the $\mathcal{R}$ data set size $N_{\mathcal{R}}$ significantly exceeds the $\mathcal{D}$ size $N_{\mathcal{D}}$, its statistical fluctuations are subdominant to those of the data set $\mathcal{D}$. The probabilistic outcome of the 2ST is thus nearly independent of the specific instance of $\mathcal{R}$ that is employed and it depends only on the level of agreement with the R distribution of the specific $\mathcal{D}$ set that is being tested, as appropriate for GoF.

Interpreting GoF as a 2ST with unbalanced samples, $N_{\mathcal{R}} > N_{\mathcal{D}}$, opens the door to the deployment of 2ST methods for GoF. Particularly relevant in our context are the classifier-based 2ST and GoF methods first proposed by Friedman in Ref. [3]. The basic idea is to train a classifier to distinguish $\mathcal{D}$ from $\mathcal{R}$. If the two data sets are drawn from the same distribution, the trained classifier will be unable to discriminate. It will however posses some discrimination power if the distributions are different. The performance of the trained classifier, as quantified by any standard classification metric, can thus be used as a metric to assess the difference between the two data sets. This procedure defines a 2ST and in turn a GoF test. More general metrics could be employed to define the GoF test, not necessarily related with classification performances. One option mentioned in Ref. [3] is to use standard univariate GoF methods, such as the Kolmogorov–Smirnov test, on the output of the trained classifier evaluated on $\mathcal{D}$ and on $\mathcal{R}$. In this case, the classifier is basically employed for dimensionality reduction and the actual GoF test is performed with a traditional univariate method.

Classifier-based GoF (C-GoF) methods have been investigated only sporadically in the high energy physics literature. See Ref. [4–6] and more recently Ref. [7]. In computer science, the simplest C-GoF implementation based on the classification accuracy metric [8] has been studied and employed quite extensively to assess the quality of generative models.

**Neyman–Pearson testing**   Hypothesis testing as formulated by Neyman and Pearson poses a third distinct statistical problem. This is connected with the GoF problem, which is in fact a type of hypothesis testing. The *null hypothesis* under examination is the reference hypothesis, $H_0 = R$. All the GoF methods proceed, like hypothesis tests do, by assigning some probabilistic measure to the data $\mathcal{D}$—such as a p-value $p[\mathcal{D}]$—that is indicative of the level of agreement of the data with their expected distribution in the R hypothesis. However, the Neyman–Pearson theory of hypothesis testing [9] also requires a second hypothesis, the alternative hypothesis $H_1$, which is however absent in the formulation of GoF problems. The alternative hypothesis plays an essential role for the design of the hypothesis test that is adequate for each specific problem. In particular, it controls the selection of the test statistic, which in turn defines the p-value. The optimal choice is the one that, at fixed type-I error rates, minimises type-II errors defined with respect to the alternative $H_1$. In essence, Neyman–Pearson testing enables a relative assessment of the $H_0 = R$ hypothesis with the data in comparison with the agreement of the alternative $H_1$ with the same data. GoF is different as it aims instead at assessing the agreement of R with the data in absolute terms.

Using Neyman–Pearson testing for GoF is straightforward, but dangerous. On one hand, a very natural pragmatic approach to assess the R distribution agreement is indeed to try and see if other distributions provide a much better fit to the data. This can be achieved by considering a deformation of the R distribution that depends on free parameters $\mathbf{w}$. This defines a family of hypotheses, i.e. a composite hypothesis $H_{\mathbf{w}}$, to be identified with the Neyman–Pearson alternative $H_1 = H_{\mathbf{w}}$. The hypothesis within the family that best fits the data, $H_{\widehat{\mathbf{w}}}$, can be compared with R as a description of the observed data. In the classical Neyman–Pearson theory, this comparison employs the likelihood ratio of the two hypotheses. On the other

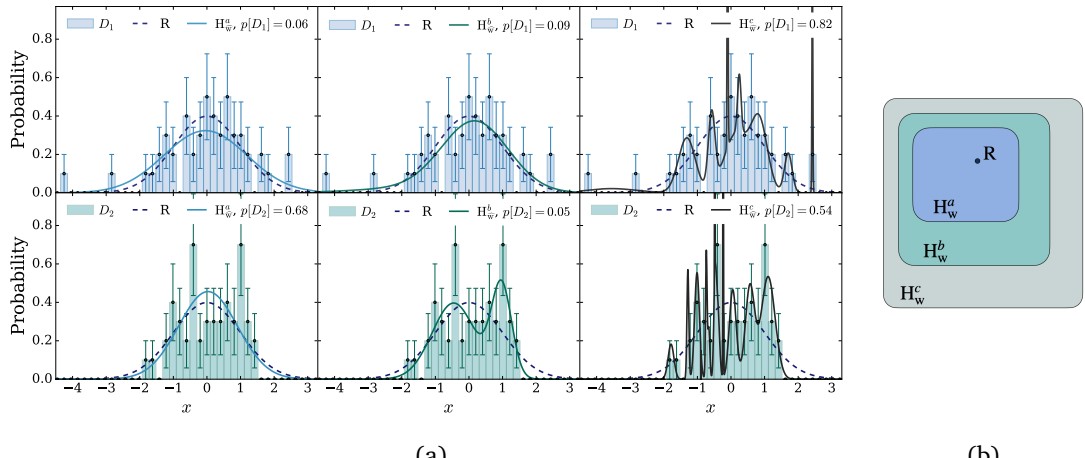

Figure 1: **One-dimensional toy problem picturing the strengths and limitations of Neyman–Pearson testing for GoF. (a)**: two datasets, $\mathcal{D}_1$ (top row) and $\mathcal{D}_2$ (bottom row) are tested to assess their compatibility with the Reference hypothesis (R), represented by the black dashed line in the left side panels. For each dataset, three Neyman–Pearson GoF tests are performed using three different alternative families: $H_{\mathbf{w}}^a$, $H_{\mathbf{w}}^b$, and $H_{\mathbf{w}}^c$. The maximum likelihood fit is showed in solid line in the left, middle and right columns respectively. The p value of each test is reported on the top right of each panel. **(b)**: illustration of the hierarchical structure of the family of hypotheses considered in the example. Larger classes of models are characterised by a higher number of Gaussian distributions in the mixture.

hand, the choice of the alternative distributions one compares against can bias the outcome of the test dramatically. As a general rule, the test will be sensitive to data departures from the R distribution only if the true data distribution is part of the $H_{\mathbf{w}}$ set, or if it is approximated reasonably well by some element of the set. The test will be instead weakly sensitive or blind to true data distributions that are outside the set.

Fig. 1a provides a practical illustration of this behaviour for one-dimensional data. Under the reference hypothesis R, data points are distributed according to a Gaussian distribution with zero mean and unit variance. We then consider three parametrised families of alternative hypotheses: $H_{\mathbf{w}}^a$, $H_{\mathbf{w}}^b$ and $H_{\mathbf{w}}^c$. Each of them defines a different Neyman–Pearson test, and in turn a different GoF method to assess the compatibility of the data with R. The upper and lower plots in the figure correspond to two different data sets $\mathcal{D}_{1,2}$ being tested, each consisting of 50 data points. The first data set $\mathcal{D}_1$ is drawn from a Gaussian with a mean of 0 and a variance of 1.2. $\mathcal{D}_2$ is sampled from a balanced mixture of two Gaussians, with means 0 and 0.83, and variances 1.2 and 0.5. The data are visualised as histograms in the plots.[1]

The first alternative $H_{\mathbf{w}}^a$ is the set of all Gaussian distributions. The best fit in this family, displayed with a solid line in the top left plot, is quite better than the standard Gaussian (dashed line) as a description of the first data set. Correspondingly, the GoF p-value that we obtain in this case is low, $p^a[\mathcal{D}_1] = 0.06$, signalling a poor agreement of the standard Gaussian hypothesis with the data. However, the best fit in $H_{\mathbf{w}}^a$ is very similar to the standard Gaussian in the case of the second data set and the corresponding p-value is high, $p^a[\mathcal{D}_2] = 0.68$. The GoF test designed as a Neyman–Pearson test with alternative $H_1 = H_{\mathbf{w}}^a$ has failed to identify the evident discrepancy between $\mathcal{D}_2$ and the reference distribution.

---

[1]The interested reader can find on https://github.com/GaiaGrosso/NPLM-GOF the straightforward implementation of the Neyman-Pearson [9] strategy we used to obtain these results.

The second alternative, $H_{\mathbf{w}}^{b}$, is an extension of $H_{\mathbf{w}}^{a}$ where two Gaussian distributions are present, with arbitrary mean and variance and arbitrary relative normalisation. The best-fit in $H_{\mathbf{w}}^{b}$ to the $\mathcal{D}_2$ data (middle-bottom plot) is very different from the standard Gaussian, and offers a much better description of the data. Consequently, $p^{a}[\mathcal{D}_2] = 0.05$ and the discrepancy of $\mathcal{D}_2$ with the R hypothesis is clearly identified if employing $H_{\mathbf{w}}^{b}$, rather than $H_{\mathbf{w}}^{a}$, for the Neyman–Pearson test. The general lesson is that the alternative $H_{\mathbf{w}}$ should be as general as possible and capable to adapt itself—namely to provide a good fit—to whatever the true data distribution is. Otherwise, the test is exposed to dramatic failures as we have seen.

The capability of $H_{\mathbf{w}}$ to fit the data accurately is not the only factor that controls the sensitivity. It is evidently easier for a more complex distribution with many free parameters to offer a better fit to the data. This fact is taken into account in the Neyman–Pearson test strategy when associating a p-value to the relative fit quality (i.e., the likelihood ratio) in the R and in the $H_{\widehat{\mathbf{w}}}$ hypotheses. The extreme situation is when the $H_{\mathbf{w}}$ distribution contains enough parameters to fit all the statistical fluctuations in the data set, or even to accommodate each individual point. In this case, the best-fit $H_{\widehat{\mathbf{w}}}$ will offer a "perfect" (but overfitted) description of the data set, much better than the one provided by the R distribution, producing a very large value for the likelihood ratio. Since statistical fluctuations are always present, this will occur for any data set and regardless of whether the true data distribution is or is not R. The Neyman–Pearson p-value is defined by comparing the observed likelihood ratio with the typical values assumed by the ratio for R-distributed data. Since both are equally large, the p-value will loose sensitivity to anomalous data sets. This behaviour is illustrated by the third alternative considered in the last column of Fig. 1a, $H_{\mathbf{w}}^{c}$. The $H_{\mathbf{w}}^{c}$ distribution, defined as the linear combination of 15 Gaussians, is too complex relative to the size $N_{\mathcal{D}} = 50$ of the data sets. The test based on $H_{\mathbf{w}}^{c}$ thus fails both on the $\mathcal{D}_1$ and on the $\mathcal{D}_2$ data sets. A successful GoF strategy based on Neyman–Pearson testing should thus balance the flexibility of $H_{\mathbf{w}}$ against the need of avoiding overfitting.

The connection between Neyman–Pearson testing and GoF was emphasised by Baker and Cousins in 1983 [10], based on literature from the 1920s. Their simple starting point was the observation that if we use the data to construct a histogram with non-overlapping bins, the counts in each bin are independent Poisson variables.[2] Independent Poisson distributions with arbitrary expected values in each bin is thus the most general possible alternative hypothesis $H_{\mathbf{w}}$. We can use this alternative for a Neyman–Pearson test assessing the agreement with the observed counts of the predictions for the expected numbers in the R hypothesis. Starting from this observation, Ref. [10] derived the $\chi^2$ approach to the GoF of binned data as a Neyman–Pearson test. The $\chi^2$ GoF with binned data is widely employed. In particular it is often used in high energy physics to assess the SM agreement with data [11]. However, binning is impractical for multi-dimensional data. Furthermore, the choice of the binning is problematic and subject to the same type of possible failures described above: too narrow bins can cause overfitting while too wide bins can be unable to accommodate important features of the data entailing sensitivity loss.

**The new phyisics learning machine (NPLM)** The authors of Ref. [12] proposed a systematic approach to the design of GoF methodologies as Neyman–Pearson tests, in which the deformation of the R distribution that defines the alternative $H_{\mathbf{w}}$ is provided by a generic parametrised family of functions $\mathcal{F} = \{f_{\mathbf{w}}(x), \forall\, \mathbf{w}\}$. In particular the functions $f_{\mathbf{w}}$ are conveniently taken to parametrise the log ratio between the $H_{\mathbf{w}}$ and R densities—see later Eq. (1). The functional set $\mathcal{F}$ could consist of neural networks [12–14], or be obtained with kernel methods [15, 16], and other options could be considered as well. Suitable regularisation strategies to avoid over-

---

[2]This holds provided the number of data points $N_{\mathcal{D}}$ is a Poisson-distributed random variable, as for natural particle physics data. The distribution would be multinomial if $N_{\mathcal{D}}$ was a fixed number.

fitting have been developed in each case [13, 15]. This Neyman–Pearson-based approach to GoF was developed for searches of New Physics in high energy collider data and it employs machine learning techniques, hence the NPLM acronym.

The performance of the NPLM method has been investigated on a number of GoF problems including toy as well as realistic new physics searches at colliders [12–15] and data quality monitoring [16]. Importantly enough, the NPLM extension to include imperfections in the knowledge of the reference hypothesis—which is straightforward in the framework of Neyman–Pearson testing—was developed and demonstrated in Ref. [14]. However, no systematic comparison has yet been performed with the many other GoF methods that exist, mainly outside the high energy physics literature. Such a comparison is the goal of the present article.

**Overview of the paper** Like any other practical GoF, the NPLM method operates as a 2ST by comparing the data $\mathcal{D}$ with a reference data set $\mathcal{R}$, as previously described. Furthermore, it entails optimising the model parameters in order to single out the best-fit to the data, $f_{\widehat{\mathbf{w}}}(x)$, by a supervised training on the $\mathcal{D}$ and $\mathcal{R}$ data sets. The whole procedure is thus very similar to the one of classifier-based approaches to GoF, as we will outline in Section 2. A comparison with C-GoF methods, with which it shares many features, is thus the natural first step for a comparative assessment of the NPLM performances. The simplest C-GoF method [8] is studied in Section 2. NPLM generally outperforms the other methods in all the benchmark problems considered in this work, and it does not show any critical failure.

The comparison is extended in Section 3 to a more general class of C-GoF tests that we design following suggestions from the literature based on the following logic. Two important features separate NPLM from existing C-GoF methods. First, the entire data set is employed for both training and evaluation. In contrast, a train-test split is more natural for a classifier-based approach. Second, the NPLM evaluation metric is provided by the log likelihood-ratio between the $H_{\widehat{\mathbf{w}}}$ and the R hypotheses, as in Neyman–Pearson testing. This quantity has no immediate interpretation as a classification metric. In Section 3, we investigate variants of the NPLM method that eliminate these peculiarities, opting for a train-test split rather than in-sample evaluation, or for standard classification metrics rather than the likelihood ratio. Following Friedman's suggestion [3], the usage of univariate GoF metrics (listed in Appendix B) is also investigated. The study of these C-GoF inspired NPLM variants reveals that both these two peculiarities of the NPLM method are beneficial for the sensitivity.

To enrich our study, we compare NPLM with traditional GoF methods on univariate data. This eliminates one of the main advantages in using machine learning models, namely their intrinsic ability to perform nonlinear dimensional reduction. The results are presented in Appendix C.

In Section 4 we summarise our findings and discuss the qualities of NPLM within the landscape of GoF methods. This is not straightforward, because the outcome of GoF comparative studies may strongly depend on the specific benchmark problems that are considered for the comparison. This risk is mitigated by employing the largest possible set of benchmarks, including those considered in previous NPLM studies [12–16] and a few new ones. While unavoidably partial and mostly inspired by high energy physics problems, our benchmarks are selected to probe the sensitivity to qualitatively different types of anomalous data. For instance, anomalies that emerge as sharp features in weakly-populated regions of the reference distribution, as opposed to departures in the bulk of the distribution. As in previous works, we ensure that anomalous data of different types are comparable by monitoring their "ideal" $Z$-score, $Z_{id}$. $Z_{id}$ is defined as the sensitivity of a hypothesis test that is fully optimised to the specific anomaly under examination. We define a "good" GoF method to be one that responds uniformly, i.e. with comparable $Z$-score, to anomalies of different type but with comparable $Z_{id}$. Qualitatively, NPLM exhibits a more uniform response than the other GoF methods consid-

ered in this paper. The notion of ideal $Z$-score, and the list of benchmark problems employed for the comparison is reviewed in Appendix A.

## 2 NPLM vs C-GoF

We start this section with a concise overview of the NPLM method [12–16], outlining its connection with C-GoF approaches [3,8]. Next, we consider the simplest C-GoF method that we introduce as a straightforward adaptation of the Classifier-based Two-Sample Test (C2ST) [8] to the case of unbalanced samples $N_{\mathcal{R}} > N_{\mathcal{D}}$. Finally, we present comparative studies of the NPLM and C2ST performances.

### 2.1 NPLM as a classifier-based GoF

The NPLM method works as follows. The alternative distribution in the $H_{\mathbf{w}}$ hypothesis of the Neyman–Pearson test—see the Introduction for an overview and Fig. 1 for an illustration—is defined as

$$n(x|H_{\mathbf{w}}) = e^{f_{\mathbf{w}}(x)} n(x|R), \tag{1}$$

in terms of a generic set of functions $\mathcal{F} = \{f_{\mathbf{w}}(x), \forall \mathbf{w}\}$, to be specified. The symbol $n$ denotes a number density distribution, namely the probability density—whose integral over $x$ is equal to 1— times the total number of points that are expected to be found in the data set $\mathcal{D}$. In particular, $n(x|R)$ denotes the distribution of the variable $x$ in the reference hypothesis R and $n(x|H_{\mathbf{w}})$ denotes the distribution under the alternative $H_{\mathbf{w}}$.

We are considering here the setup, typical of high energy physics, where the number of points in $\mathcal{D}$, $N_{\mathcal{D}}$, is itself a random variable, and follows the Poisson distribution. The mean of this Poisson distribution in the R hypothesis, denoted as $N(R)$, is the integral of $n(x|R)$. The expectation in the $H_{\mathbf{w}}$ hypothesis, $N(H_{\mathbf{w}})$, is the integral of $n(x|H_{\mathbf{w}})$ as defined by Eq. (1). Hence, $N(H_{\mathbf{w}})$ depends on the function $f_{\mathbf{w}}$ and thus on the alternative hypothesis parameters $\mathbf{w}$.

While introduced in the setup with variable Poisson-distributed $N_{\mathcal{D}}$, the NPLM method is perfectly suited, and will be employed in some of the studies that follow, to deal with the cases in which the number of measured data points $N_{\mathcal{D}}$ is not a random variable. In these cases, one simply replaces $N(R)$ with $N_{\mathcal{D}}$ in all the equations that follow.

The model $f_{\mathbf{w}}(x)$ needs to be trained in order to identify the specific distribution, $n(x|H_{\widehat{\mathbf{w}}})$, that best fits the observed data. Training is performed on the observed data set $\mathcal{D}$, labelled as $y = 1$, and on the reference data $\mathcal{R}$ labelled as $y = 0$. Training exploits a classical result of statistical learning: a continuous-output classifier trained to tell apart two sets of data approximates—possibly up to a given monotonic transformation—the log ratio of their population distributions. A suitable loss function, for which this property is proven explicitly in, e.g., Ref. [12,15], is the weighted logistic loss

$$\ell(y, f_{\mathbf{w}}(x)) = (1 - y)\frac{N(R)}{N_{\mathcal{R}}} \log\left[1 + e^{f_{\mathbf{w}}(x)}\right] + y \log\left[1 + e^{-f_{\mathbf{w}}(x)}\right]. \tag{2}$$

An alternative loss with the same property is the maximum-likelihood loss [12,14]

$$\ell(y, f_{\mathbf{w}}(x)) = (1 - y)\frac{N(R)}{N_{\mathcal{R}}}\left(e^{f_{\mathbf{w}}(x)} - 1\right) - y f_{\mathbf{w}}(x). \tag{3}$$

The maximum-likelihood loss is a more natural choice in the context of Neyman–Pearson testing, because its minimisation is equivalent to the maximisation of the likelihood in the $N_{\mathcal{R}} \to \infty$ limit [12,15]. In this way, the best-fit $n(x|H_{\widehat{\mathbf{w}}})$ obtained with this loss coincides

with the maximum-likelihood best-fit that is employed in the classical Neyman–Pearson theory [9].

After training, NPLM proceeds as a Neyman–Pearson test by evaluating a test statistic that is twice the logarithm of the likelihood ratio in the $H_{\widehat{\mathbf{w}}}$ and in the R hypotheses. The relevant likelihood for our problem is the extended likelihood

$$\mathcal{L}(\mathrm{H}|\mathcal{D}) = \frac{e^{-\mathrm{N(H)}}}{\mathrm{N}_{\mathcal{D}}!} \prod_{x \in \mathcal{D}} n(x|\mathrm{H}), \tag{4}$$

for a generic hypothesis H. Therefore, using Eq. (1)

$$-2\log\frac{\mathcal{L}(\mathrm{R}|\mathcal{D})}{\mathcal{L}(\mathrm{H}_{\widehat{\mathbf{w}}}|\mathcal{D})} = -2\left[\mathrm{N}(\mathrm{H}_{\widehat{\mathbf{w}}}) - \mathrm{N}(\mathrm{R}) - \sum_{x \in \mathcal{D}} f_{\widehat{\mathbf{w}}}(x)\right]. \tag{5}$$

Recalling that $\mathrm{N}(\mathrm{H}_{\mathbf{w}})$ is the integral of $n(x|\mathrm{H}_{\mathbf{w}})$ in Eq. (1), and approximating the integral with a Monte Carlo sum over the reference sample $\mathcal{R}$, we finally obtain the Likelihood Ratio (LR) test statistic that is employed in NPLM

$$t_{\mathrm{LR}} = -2\left[\frac{\mathrm{N(R)}}{\mathrm{N}_{\mathcal{R}}} \sum_{x \in \mathcal{R}} (e^{f_{\widehat{\mathbf{w}}}(x)} - 1) - \sum_{x \in \mathcal{D}} f_{\widehat{\mathbf{w}}}(x)\right]. \tag{6}$$

Large values of $t_{\mathrm{LR}}$ signal that $H_{\widehat{\mathbf{w}}}$ offers a better description of the data than R, as the likelihood is larger. In turn, this disfavours R being the true distribution of the data.

Like for any hypothesis test, or two-sample test, the value of the test statistic is not an indicator of the data agreement with the R hypothesis under examination in absolute terms, but only in comparison with the typical values it assumes when the data are truly distributed according to R. A proper probabilistic indicator of the compatibility of the data with R is the p-value

$$\mathrm{p}[t] = \int_t^\infty dt' \, p(t'|\mathrm{R}), \tag{7}$$

which accounts for the probability distribution of the test statistic in the R hypothesis, $p(t|\mathrm{R})$. In some cases, $p(t|\mathrm{R})$ can be estimated analytically. This is not the case in NPLM, and $p(t|\mathrm{R})$ is computed empirically by employing artificial sets of data—called *toy data*—that follow the R distribution by construction. The toy data sets are built out of R-distributed data points, different from those employed to form the reference sample $\mathcal{R}$.

It is worth noting that the test statistic (6) features both an explicit dependence on the data $\mathcal{D}$—in the second summation—and an implicit dependence from the fact that the best-fit model $f_{\widehat{\mathbf{w}}}(x)$ does depend on the data set $\mathcal{D}$, which is used for training. In order to compute $p(t|\mathrm{R})$ we thus need to first train the model, and next evaluate $t_{\mathrm{LR}}$, on each toy data set. Also notice that $t_{\mathrm{LR}}$ depends, both explicitly and implicitly, also on the reference data set $\mathcal{R}$. We explained in the Introduction that it is expected—and can be verified—that the dependence on the $\mathcal{R}$ set is weak in the unbalanced limit $\mathrm{N}_{\mathcal{R}} \gg \mathrm{N}_{\mathcal{D}}$, as the $\mathcal{R}$ set provides in this limit a nearly perfect representation of the R hypothesis distribution. Nevertheless, the statistical fluctuations of the $\mathcal{R}$ set are taken into account in our evaluation of $p(t|\mathrm{R})$ by employing toy data sets also for the reference sample.

There are currently two implementations of NPLM, where the model $f_{\mathbf{w}}(x)$ is respectively a neural network (NPLM-NN [12–14]), or it is built with kernel methods (NPLM-KM [15,16]). Each implementation comes with a dedicated prescription for the selection of the model, training and regularisation hyper-parameters. These prescriptions form an integral part of the NPLM method as they ensure the required balance between the models' flexibility and the

need of avoiding overfitting. The selected hyper-parameters depend in general on the expected data size N(R), on the reference size $N_{\mathcal{R}}$, on the dimensionality of the variable $x$ and ultimately on its distribution in the R hypothesis. Therefore, the hyper-parameters need to be selected for the NPLM application to each given GoF setup. On the other hand, the hyper-parameters selection does not depend on and it is not optimised for the detection of any specific type of data departure from the reference hypothesis, as appropriate for a GoF method. After the hyper-parameters are selected for a given reference hypothesis, the GoF algorithm must run identically on all the data sets that are employed for testing its sensitivity to anomalous data and no a posteriori re-optimisation is allowed.

The comparisons performed so far—in particular in Ref. [15]—did not reveal a major difference in performance between the NPLM-NN and the NPLM-KM implementations, suggesting that the specific model employed for $f_{\mathbf{w}}(x)$ is not the key factor controlling the sensitivity. Furthermore, since NPLM-NN employs the maximum-likelihood loss (3) while NPLM-KM uses the weighted logistic loss (2), the choice of the loss function is also expected to play a minor role. Several of the benchmark problems of the present paper have been studied both with NPLM-NN and with NPLM-KM, obtaining similar performances.

The NPLM method can be interpreted as a classifier-based method, if we follow the general notion of C-GoF given by Friedman in Ref. [3]. In fact, a general C-GoF is any algorithm that performs the three following operations. First, training a classifier between $\mathcal{D}$ and $\mathcal{R}$. The NPLM model $f_{\mathbf{w}}(x)$ is in fact a continuous-output classifier that we train between $\mathcal{D}$ and $\mathcal{R}$. The correspondence can be made more explicit by defining a classification function $c_{\mathbf{w}}(x) \in [0, 1]$ out of $f_{\mathbf{w}}(x) \in \mathbb{R}$ by the monotonic transformation

$$c_{\mathbf{w}}(x) = \frac{1}{1 + e^{-f_{\mathbf{w}}(x)}}. \tag{8}$$

Notice that the weighted logistic loss (2) reduces to the weighted binary cross-entropy after this transformation. Second, evaluating the trained classifier on $\mathcal{D}$ and $\mathcal{R}$. This is what NPLM does on the right hand side of Eq. (6). The third step is to define and compute, on the evaluated classifier, some test statistic that is sensitive to the discriminating power of the classifier between the two sets. The $t_{\mathrm{LR}}$ test statistics (6) is definitely not a standard metric of classification. But nevertheless it is preferentially large it the $f_{\widehat{\mathbf{w}}}$ function is large and positive (i.e., $c_{\widehat{\mathbf{w}}} \to 1$) on the $\mathcal{D}$ set and large and negative (i.e., $c_{\widehat{\mathbf{w}}} \to 0$) on the $\mathcal{R}$ set. Therefore, it is indicative of the $f_{\widehat{\mathbf{w}}}$ ability to tell $\mathcal{D}$ from $\mathcal{R}$ and as such it can be used as a classification metric.

In spite of this formal correspondence with C-GoF, NPLM is a very different approach. Its peculiarities including the choice of the test statistics stem from its origin as a Neyman–Pearson test and are unrelated with the theory of classification. This is effectively illustrated by comparing NPLM with the simplest C-GoF method that we review in the following section. We will come back in Section 3 to the discussion of the characteristics of NPLM in the landscape of C-GoF tests.

## 2.2 C2ST with unbalanced samples

C2ST [8] was originally formulated as a 2ST with balanced data sets, $N_{\mathcal{R}} = N_{\mathcal{D}}$. We first discuss it in this configuration before introducing its straightforward adaptation to the unbalanced case $N_{\mathcal{R}} > N_{\mathcal{D}}$. The number of observations $N_{\mathcal{D}}$ (and $N_{\mathcal{R}}$) is a pre-specified fixed number. We will consider the case in which it fluctuates as a Poisson variable in the next section.

The first step is to split the $\mathcal{D}$ and $\mathcal{R}$ sets in two equal parts, obtaining two pairs of samples $(\mathcal{D}_{tr}, \mathcal{R}_{tr})$ and $(\mathcal{D}_{te}, \mathcal{R}_{te})$ to be used for training and for testing, respectively. Each of the four sets contains $N_{\mathcal{D}}/2$ points. This training-test splitting is very natural in the classifier-based context. The aim there is to probe the dissimilarity of $\mathcal{D}$ to $\mathcal{R}$ by assessing the performances

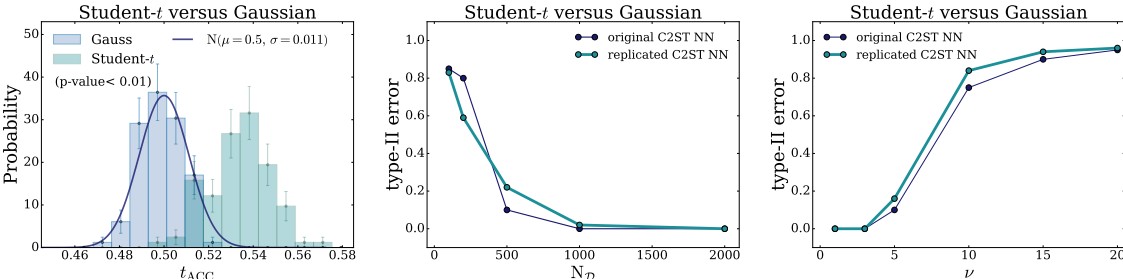

Figure 2: **C2ST.** Left panel: $t_{ACC}$ distributions in two different hypotheses for the underlying data distributions. Central panel: Type-II error at 95% confidence level varying $N_{\mathcal{D}}$, with $\nu$ set at 3. Right panel: Type-II error at 95% confidence level varying $\nu$ with $N_{\mathcal{D}} = 2000$.

that a classifier can attain in distinguishing them. Test data that are independent from the training data are evidently needed for this assessment.

Again very naturally from the viewpoint of classification, the binary accuracy of the trained classifier, with a threshold of $1/2$, is used as test statistics

$$t_{ACC} = \frac{1}{N_{\mathcal{D}}}\left[\sum_{x \in \mathcal{R}_{te}} \mathbb{I}[c_{\widehat{\mathbf{w}}}(x) < 1/2] + \sum_{x \in \mathcal{D}_{te}} \mathbb{I}[c_{\widehat{\mathbf{w}}}(x) > 1/2]\right], \qquad (9)$$

where $\mathbb{I}$ denotes the indicator function.

We implemented the C2ST method in the simplest setup considered in Ref. [8], where the variable $x$ is one-dimensional and the R hypothesis for its distribution is the standard Gaussian. The number of data points is fixed to $N_{\mathcal{D}} = N_{\mathcal{R}} = 2000$ if not specified otherwise. Smaller data size setups have been also considered with 100, 200, 500, or 1000 data points. As in [8], we employ a neural network with architecture 1-20-1, Adam optimizer, binary cross-entropy loss function and 100 training epochs. As compared with the other models studied in [8], this is among those that offer the best performances on the one-dimensional problem under examination.

We generate, according to the standard Gaussian hypothesis R, 100 toy instances of the $\mathcal{D}$ and of the $\mathcal{R}$ data sets. For each instance, training is performed on half of the data and the $t_{ACC}$ test statistics (9) is evaluated on the remaining data. The resulting $p(t_{ACC}|R)$ distribution is displayed on the left panel of Fig. 2 (light blue histogram). As noticed in [8], $p(t_{ACC}|R)$ is well-approximated by a Gaussian with a mean of 0.5 and a standard deviation of $1/(2\sqrt{N_{\mathcal{D}}}) = 0.011$. Using this distribution we can associate a p-value to the observed value of $t_{ACC}$ by means of Eq. (7).

As in [8], we evaluate the method performances to detect anomalous data sets $\mathcal{D}$ that follow a Student-t distribution rather than the standard Gaussian. The Student-t is characterised by the number of degrees of freedom, $\nu$, and for larger $\nu$ it approaches the standard Gaussian making it increasingly difficult to detect the anomaly. The $t_{ACC}$ distribution on 100 $\mathcal{D}$ sets drawn from the Student-t with $\nu = 3$—while the $\mathcal{R}$ sets are, of course, still drawn from the standard Gaussian—is displayed on the left panel of Fig. 2 (light green histogram). As the distribution is quite different from the one observed in the R hypothesis, the test possesses good discriminating power. The median p-value is found to be below what can be quantified empirically with 100 R-distributed toys, so below around 0.01.

In light of some confusion that occasionally emerges in the literature (see e.g. [17,18]) on the usage of classifier-based tests, it is worth to emphasize that the classification accuracy—or its complement, the misclassification error—should not be confused with the p-value or any

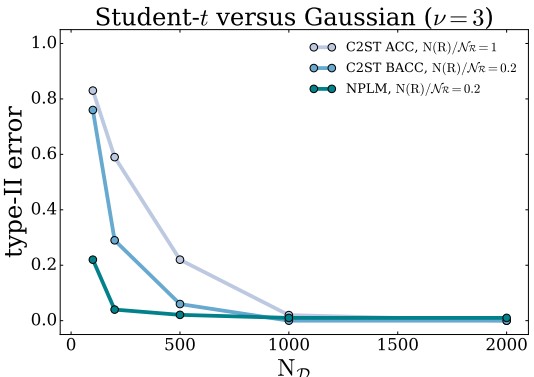
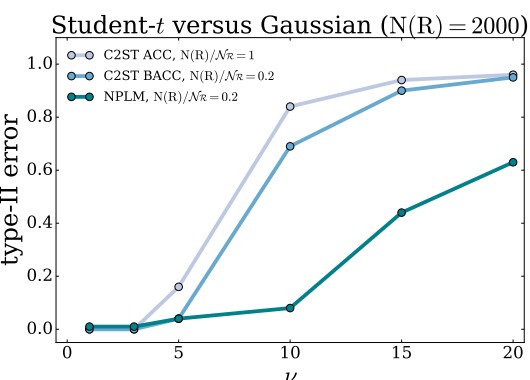

Figure 3: **C2ST vs. NPLM.** Same as Fig. 2, comparing the C2ST method with balanced samples (BACC), the one with unbalanced samples (ACC) with $N_{\mathcal{R}}/N_{\mathcal{D}} = 5$, and the NPLM method in its default implementation based on maximum-likelihood-ratio.

other probabilistic indicator of the data agreement with the R hypothesis. The accuracy will be typically poor, and close to the random classifier accuracy of 0.5. The discrimination power of the test emerges from relatively small departures of $t_{\mathrm{ACC}}$ from 0.5, which are however highly unlikely to occur for R-distributed data. The left panel of Fig. 2 displays this typical behaviour.

The central and right panels of Fig. 2 display the performances of our implementation of C2ST and their agreement with the original results in [8]. The plots show the type-II error of the test at 95% confidence level. Namely, we set a threshold p = 0.05 on the p-value, below which we label the R hypothesis as excluded. Next we compute the probability of not excluding R when the data are in fact not distributed according to R, but to one of the alternatives.[3] As expected, the probability of type-II error decreases with $N_{\mathcal{D}}$ (central panel) as larger data sets possess more discriminating power. It increases with $\nu$ (right panel) as the Student-t distribution approaches the standard Gaussian.

The C2ST method is easily extended to the case of unbalanced samples $N_{\mathcal{R}} > N_{\mathcal{D}}$. Training/test splitting is performed in equal portions as before. The loss function is the weighted cross-entropy—specifically, the later Eq. (11) with N(R) replaced by $N_{\mathcal{D}}$. The test statistic is the balanced classification accuracy

$$t_{\mathrm{BACC}} = \frac{1}{N_{\mathcal{R}}} \sum_{x \in \mathcal{R}_{te}} \mathbb{I}[c_{\widehat{\mathbf{w}}}(x) < 1/2] + \frac{1}{N_{\mathcal{D}}} \sum_{x \in \mathcal{D}_{te}} \mathbb{I}[c_{\widehat{\mathbf{w}}}(x) > 1/2]. \qquad (10)$$

We tested this version of the C2ST strategy in the same setup above, but raising $N_{\mathcal{R}}$ to 5 times $N_{\mathcal{D}}$. We employ the same neural network model, but we notice that more training epochs are needed for convergence owing to the larger training set. We employ 500 epochs apart from the setup with $N_{\mathcal{D}} = 100$, where 100 epochs are used. The number of epochs has been selected by running training on a few R-distributed toy data sets, and monitoring the evolution of the balanced accuracy during training. We selected the number of epochs at which the accuracy on a validation sample stopped improving.

The results are displayed in Fig. 3. As expected, C2ST with unbalanced samples is more effective than the balanced one because it exploits the larger statistics that is available in the $\mathcal{R}$ sample. Notice that performances do not improve indefinitely increasing $N_{\mathcal{R}}$ at fixed $N_{\mathcal{D}}$. As soon as $N_{\mathcal{R}}$ exceeds $N_{\mathcal{D}}$ by a factor of few, it offers a description of the R distribution that is "perfect", in comparison with the description of the true distribution that is offered by the

---

[3]The results are presented in this form for a direct comparison with Ref. [8]. No hard exclusion threshold on the p-value will be employed in the rest of the paper.

data $\mathcal{D}$. Therefore the performances quickly saturate and no significant gain will be observed if raising $N_{\mathcal{R}}/N_{\mathcal{D}}$ above 5. The figure also displays that an even more significant performance gain is attained with the NPLM method, in the same setup with $N_{\mathcal{R}}/N_{\mathcal{D}} = 5$. This is discussed in the following section.

## 2.3 Performance comparison

For a first comparative assessment of the performance, we applied NPLM to the Gaussian vs Student-t discrimination problem. Specifically, we employed the NPLM-NN implementation based on neural networks. The hyper-parameters are selected on the basis of the $\chi^2$-compatibility criterion of Refs. [13, 14], focusing on a simple neural network model $(1, 3, 1)$. With the Gaussian reference distribution, $N_{\mathcal{R}}/N_{\mathcal{D}} = 5$, and for $N_{\mathcal{D}} = 100, 200, 500, 1000$ and 2000, this criterion results in weight clipping regularisation parameters of 6, 5.2, 5, 4.7 and 4.2, respectively. 100000 training epochs ensure convergence in all configurations. After this choice of the hyper-parameters is performed, based exclusively on $\chi^2$-compatibility in the R hypothesis and with no reference to any alternative distribution to be later employed to test the performances, the method is applied to the anomalous data sets generated with the Student-t distribution.

The NPLM results, reported in Fig. 3, are significantly better than the ones of C2ST. The possible origin of the improved NPLM performances, on these and other benchmarks considered below, is discussed briefly at the end of this section. In Section 3 we analyse in detail the main methodological differences between C2ST and NPLM and assess their impact on the performances.

We now turn to the benchmark GoF problems defined in Appendix A. These are mostly inspired by and representative of use cases that are encountered in high energy collider physics, hence the total number of points in the $\mathcal{D}$ data set, $N_{\mathcal{D}}$, is a Poisson-distributed random variable as explained in Section 2.1. However, the C2ST method is not ideally suited to deal with this setup. Before comparing it with NPLM, we thus need to introduce one last improvement of the C2ST methodology.

If $N_{\mathcal{D}}$ is a random variable and not a pre-specified number, its observed value carries information on the viability of the R hypothesis. The expected value of $N_{\mathcal{D}}$ in the R hypothesis is $N(R)$. Hence in particular if $N_{\mathcal{D}}$ departs from $N(R)$ more than typically expected from Poisson fluctuations, this very fact signals that the data are in tension with R even if their probability distribution was identical to the one predicted by the R hypothesis. The plain C2ST method is not suited to exploit $N_{\mathcal{D}}$ as a discriminating variable, because of two issues.

The first issue is that the regular weighted binary cross-entropy loss function is specifically designed to be insensitive to the size of the two training samples, hence in particular to $N_{\mathcal{D}}$. The trained classifier resulting from the optimisation of the loss will depend only on the probability distribution of the two classes. In particular in the extreme case where the data probability distribution is identical to the reference one, the trained classifier function will be close to the non-decision boundary of $1/2$ and retain no information on the possible departure of $N_{\mathcal{D}}$ from $N(R)$. This is easily remedied by weighting the loss function with $N(R)$ in place of $N_{\mathcal{D}}$, namely using

$$\ell(y, c_{\mathbf{w}}(x)) = -(1-y)\frac{N(R)}{N_{\mathcal{R}}}\log[1-c_{\mathbf{w}}(x)] - y\log[c_{\mathbf{w}}(x)]. \tag{11}$$

Notice that this loss function is identical to Eq. (2) after trading $f_{\mathbf{w}}$ for $c_{\mathbf{w}}$ by Eq. (8).

Unlike the regular weighted cross-entropy, which employs $N_{\mathcal{D}}$ in place of $N(R)$, the minimisation of Eq. (11) is sensitive to $N_{\mathcal{D}}$. In particular, large $N_{\mathcal{D}}$ will boost the importance of the second term of the loss function, which will be evaluated on a larger $\mathcal{D}$ set. The trained classifier function $c_{\widehat{\mathbf{w}}}$ will be thus pushed towards 1 if $N_{\mathcal{D}}$ is large. It will be conversely pushed

towards 0 if $N_{\mathcal{D}}$ is small.[4]

The second issue is that the C2ST test statistics (10) does not respond well to $c_{\widehat{\mathbf{w}}}(x)$ classifiers that are systematically larger or smaller than the threshold of 1/2. If the $c_{\widehat{\mathbf{w}}}(x)$ is either always smaller or always larger than 1/2, $t_{\text{BACC}}$ is equal to the random classifier accuracy of 0.5. A test based on the $t_{\text{BACC}}$ test statistics is thus insensitive to departures of $N_{\mathcal{D}}$ from N(R), because their effect on the trained classifier is precisely to push it above or below 1/2 uniformly in the $x$ space. We thus consider a modified version of the balanced accuracy

$$t'_{\text{BACC}} = \frac{2}{\text{N(R)} + N_{\mathcal{D}}} \left[ \frac{\text{N(R)}}{\text{N}_{\mathcal{R}}} \sum_{x \in \mathcal{R}_{te}} \mathbb{I}[c_{\widehat{\mathbf{w}}}(x) < 1/2] + \sum_{x \in \mathcal{D}_{te}} \mathbb{I}[c_{\widehat{\mathbf{w}}}(x) > 1/2] \right] . \tag{12}$$

In the limiting case $N_{\mathcal{D}} \gg \text{N(R)}$, such that $c_{\widehat{\mathbf{w}}}$ is pushed above 1/2 everywhere, the modified accuracy equals $t'_{\text{BACC}} = N_{\mathcal{D}}/(N_{\mathcal{D}} + \text{N(R)}) > 0.5$, while if $N_{\mathcal{D}} \ll \text{N(R)}$ and $c_{\widehat{\mathbf{w}}} < 1/2$, $t'_{\text{BACC}} = \text{N(R)}/(N_{\mathcal{D}} + \text{N(R)}) > 0.5$. In both cases the test statistics will thus assume an anomalously large value, above the indecisive classifier threshold of 0.5, offering sensitivity to the anomalous observed value of $N_{\mathcal{D}}$.

The modified C2ST method based on $t'_{\text{BACC}}$, and employing the loss function in Eq. (11), has been found to perform better than the regular C2ST on the benchmark problems we studied, and in particular as expected on those data sets where $N_{\mathcal{D}}$ departures from N(R) are statistically significant. The modified C2ST results are thus used for a fair comparison with the NPLM performances.

Among the benchmark GoF problems of Appendix A, we consider those with 1D exponential reference distribution (Expo) and the 5D ones with di-muon final states with invariant mass cuts at 60 and at 100 GeV ($\mu\mu$-60 and $\mu\mu$-100). For C2ST we employ a 1-20-1 network and a 5-20-1 network for the 1D and 5D setups, respectively. We use the Adam optimiser, 500 training epochs in 1D and 3000 epochs in the 5D setup. The number of epochs is selected with the criterion explained at the end of Section 2.2.

The NPLM performances on the benchmark problems are illustrated by the NPLM implementation based on neural networks. The hyper-parameter selection is performed with the standard NPLM-NN strategy and the selected hyper-parameters are reported in Appendix A. Most of the benchmark problems we consider here have been investigated already in previous works and the performances of NPLM-NN compared and found similar to those of the NPLM-KM implementation that employs kernel methods [15]. This is confirmed by the NPLM-KM results on the same benchmarks reported in the following. See for instance Fig.s 11 and 12.

The results are presented—in Fig.s 4 and 5—by plotting the test power as a function of its significance (or size) $Z_\alpha$. These power curves are obtained as follows. The p-value returned by the GoF test on each data sample is converted into a significance $Z$-score by the definition

$$\text{p}[Z] = \int_Z^\infty dZ' g(Z') = 1 - G[Z] \;\Rightarrow\; Z[\text{p}] = G^{-1}[1 - \text{p}], \tag{13}$$

where $g$ is the standard Gaussian pdf and $G^{-1}$ the inverse of its cumulative. With this definition, a significance of $Z = 2$ (or, of $2\sigma$) corresponds to a p-value of around 2.3%. The power curve is the probability of obtaining on data a $Z$-score that is larger than a given threshold $Z_\alpha$, i.e., a p-value that is smaller than the corresponding $\text{p}[Z_\alpha]$. This probability, $p(Z > Z_\alpha | \text{H}_{\text{T}})$, can be evaluated under different hypotheses $\text{H}_{\text{T}}$ for the true data distribution in order to test the ability of the GoF test to spot out data departures from the R hypothesis. Notice that if the

---

[4]This does not happen with the regular weighted cross-entropy because the increase (for large $N_{\mathcal{D}}$) of the second term is accompanied by the increase of the first one due to its prefactor, which is proportional to $N_{\mathcal{D}}$ in the regular cross-entropy and not set to the fixed value of N(R).

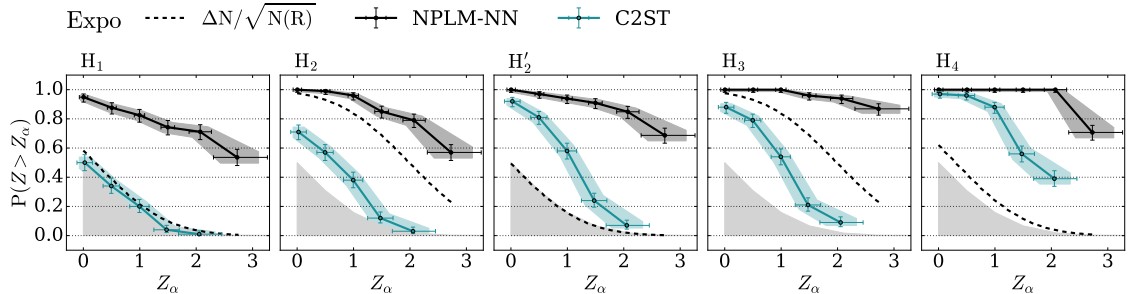

Figure 4: Power curves for C2ST and NPLM in the five signal benchmarks considered for the 1D exponential setup. The sensitivity of the event-counting test ($\Delta N/\sqrt{N(R)}$) is also shown.

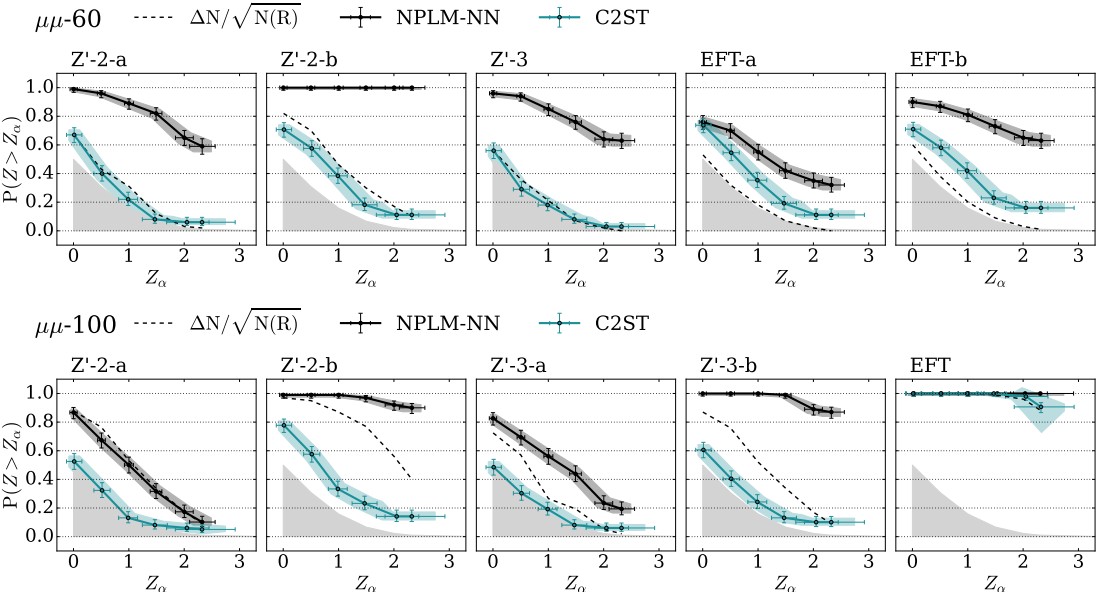

Figure 5: Power curves for C2ST and NPLM in the five signal benchmarks considered for the $\mu\mu$-60 and $\mu\mu$-100 setups. The sensitivity of the counting test ($\Delta N/\sqrt{N(R)}$) is also shown.

true hypothesis is R, $p(Z > Z_\alpha|R) = p[Z_\alpha]$ by definition. This curve is the contour of the grey region in the plots. It represents the power curve of the test under the R hypothesis. A GoF method can be claimed to be sensitive to a certain alternative $H_T \neq R$ only if the corresponding power curve is well above the grey region, with higher curves indicating better performances.

The NPLM power curves are always above those obtained with C2ST, signalling universally better performances. Particularly striking are the results obtained in those configurations where the anomalous data behaviour is due to the presence of a few signal events that emerge in a weakly-populated region of the reference distribution. For instance, in the $H_1$ benchmark—see Appendix A and in particular Fig. 8—the signal consists of an average of $S = 10$ events on top of $B = 2000$ background events that follow the reference distribution. This small S/B ratio can be sufficient to spot out the anomalous nature of the data, because the signal events are untypical (specifically, $x$ is large) in the R hypothesis distribution. In fact, the NPLM method displays good sensitivity. C2ST is instead completely blind to the 1D $H_1$ signal, with a power curve that is right on top of the reference hypothesis curve. The same behaviour is observed in those 5D problems that similarly display a small S/B.

The C2ST insensitivity to this type of configurations can be understood as follows. The trained classifier has a chance to feature a good discriminating power between the two samples only in those regions of $x$ where there is a discrepancy between the true and the reference data distribution. However, in the non-discrepant regions the trained classifier function oscillates around $1/2$ and classifies $\mathcal{D}$ and $\mathcal{R}$ data points randomly. The number of accurately classified points in the latter region will be around a half of the number of points in that region, with fluctuations of the order of the square root of the number of points. If this number is overly large in comparison with the number of points that fall instead in the discrepant region, the good classification performances in that region are overwhelmed by the statistical fluctuation in the non-discrepant region. The classification accuracy does not display an anomalously large value and the anomalous nature of the data set cannot be detected.

The situation is radically different in NPLM because the NPLM test statistics $t_{\text{LR}}$ in Eq. (6) weights individual points in the $\mathcal{D}$ set by the value assumed by the $f_{\widehat{\mathbf{w}}}(x)$ function on that point, unlike the accuracy used by C2ST that weights all the well-classified points the same. Recalling Eq. (8), we see that $f_{\widehat{\mathbf{w}}}$ is close to zero when $c_{\widehat{\mathbf{w}}} \simeq 1/2$, corresponding to an indecisive classifier. On the contrary, $f_{\widehat{\mathbf{w}}}$ is large in absolute value for confident classification $c_{\widehat{\mathbf{w}}} \simeq 0$ or $c_{\widehat{\mathbf{w}}} \simeq 1$. This mitigates the contribution from the non-discrepant regions and enhances the one from the discrepant regions, where $f_{\widehat{\mathbf{w}}}$ is large. This inherent virtue of the usage of the likelihood ratio test statistic in NPLM was emphasised already in Ref. [12].

The C2ST method displays some sensitivity to those anomalies that emerge from a distortion of the reference distribution that is less localised in the $x$ space, but still its performances are way inferior to the ones of NPLM. Often, C2ST is even less sensitive than a basic GoF strategy that is merely based on counting of the total number of observed events. The power curve of the counting test is displayed as dashed lines in the plots. The test is based on the test statistic $\Delta N/\sqrt{N(R)}$, where $\Delta N = |N_{\mathcal{D}} - N(R)|$, and thus it is only sensitive to departures of $N_{\mathcal{D}}$ from the expected value $N(R)$. Therefore it is completely insensitive to those setups where the expected value of $N_{\mathcal{D}}$ in the alternative hypothesis is identical to $N(R)$, and very weakly sensitive to small S/B setups. The fact that C2ST is often less effective than the counting test shows that for C2ST it is still difficult to exploit the discriminating power of $N_{\mathcal{D}}$ even in its improved version.

## 3  Classifier-inspired NPLM variants

The NPLM advantages observed in the direct comparison with C2ST must be due to some aspects of the NPLM method which depart strongly from the classifier-based approach that underlies the C2ST method design. In this section we identify these peculiarities and assess their impact by studying the performances of some variants of the NPLM strategy.

We emphasised in Section 2.1 that, while NPLM formally belongs to the general family of classifier-based methods as defined by Friedman in Ref. [3], it displays a number of peculiarities that stem from its origin as a Neyman–Pearson test. One of these peculiarities is definitely the usage of the maximum-likelihood loss in Eq. (3). However, we argued that this can not be responsible for the good NPLM performances because the NPLM implementation based on kernel methods employs the more standard logistic loss function in Eq. (2) and it is as effective as the implementation with neural networks that uses maximum-likelihood. Unpublished tests performed with neural networks and logistic loss, mentioned in Ref. [12], further support the claim that the maximum-likelihood loss is not essential for the NPLM performances.

The models used in the two NPLM implementations (neural networks and kernels) are routinely employed as classifiers and therefore their choice is not a peculiarity of the NPLM method. What is instead peculiar is the NPLM selection of the model hyper-parameters in-

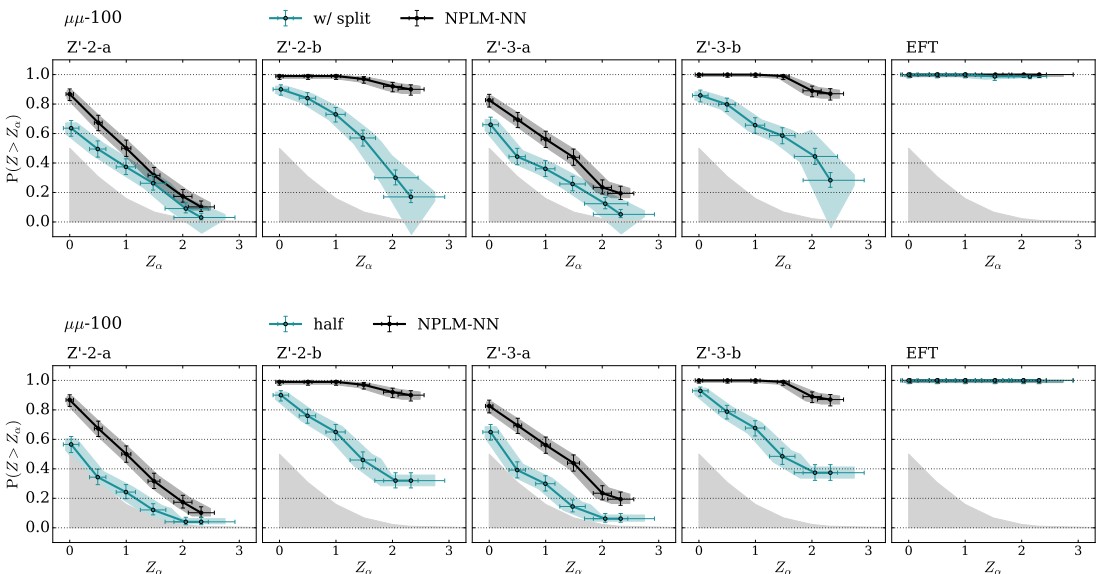

Figure 6: Power curves in the train-test split configuration (first row of plots) and in the default NPLM configurations but using only half of the data statistics (second row). The baseline NPLM results are shown for comparison.

cluding regularisation, and this aspect could in principle play an important role. On the other hand, the NPLM hyper-parameters selection protocol in essence just aims at preventing overfitting to occur on R-distributed toy data, a criterion that would be reasonable also from a classifier-based perspective. Therefore, studying the impact of the hyper-parameters on the performances is not directly relevant for the NPLM comparison with classifier-based methods.

In what follows, we will then focus on two remaining unique aspects of NPLM: the in-sample evaluation of the classifier without train-test splitting and the usage of the likelihood ratio as test statistics. The impact on the performances of these two choices are discussed in the next two sections in turn.

## 3.1 Train-test splitting

Estimating the performances of a classifier requires test data that are independent from the data used for training. The splitting of the $\mathcal{D}$ and $\mathcal{R}$ data sets into training and test is thus a very natural design choice in the classifier-based approach as previously emphasised in Section 2.2. Equally naturally, no such splitting should be performed from the viewpoint of a Neyman–Pearson test. The aim there is to assess how well the data are described by the distribution that best fits them within a certain family. Therefore, the fit (i.e., the training) must be performed with the very same data used to test. In NPLM, the test statistics—specifically, the likelihood ratio test statistics in Eq. (6)—is thus evaluated on the entire sets of available $\mathcal{D}$ and $\mathcal{R}$ data that are also employed for training.

In order to assess the impact of this in-sample evaluation on the NPLM performances, we consider a variant of NPLM in which the $\mathcal{D}$ set is first split into two equal parts, obtaining a set $\mathcal{D}_{tr}$ that is employed for training and an independent set, $\mathcal{D}_{te}$, on which the test statistics is evaluated. The same $\mathcal{R}$ is used for training and testing.[5] Other than that, we proceed like in baseline NPLM using the likelihood ratio test statistic and selecting the model, training and

---

[5]We could have decided to split $\mathcal{R}$ as well, but it is expected to result in a comparable performance because for $N_{\mathcal{R}} > N_{\mathcal{D}}$ the $\mathcal{R}$ set can be regarded as a nearly perfect description of the R distribution. The result of training and the value of $t_{\mathrm{LR}}$ are thus independent of the specific instance of the $\mathcal{R}$ set that is employed.

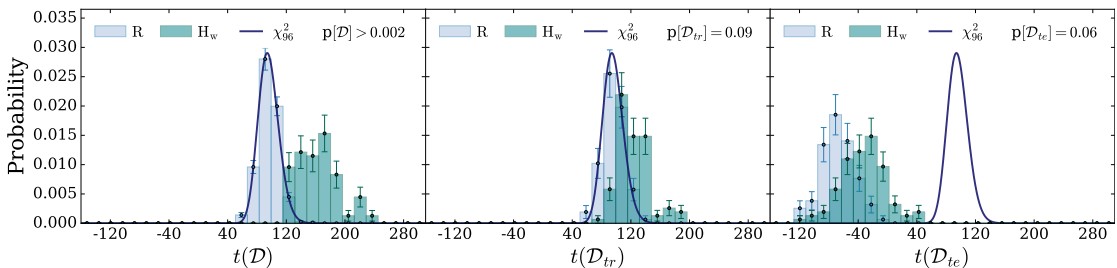

Figure 7: Test statistics distributions in the reference hypothesis and in one of the alternatives for the baseline NPLM method employing the full data statistics (left), the one employing only half of the data (center) and the NLPM variant with train-test splitting (right).

regularisation hyper-parameters to avoid overfitting on R-distributed data sets. Notice that the selected parameters are different (see Appendix A) than the ones of regular NPLM method applied on the same setup but without the train-test splitting, because training is performed in the latter case with the total $\mathcal{D}$ statistics while only half of the statistics is employed here due to the splitting.

The results are displayed in Fig. 6 in terms of the power curves defined and computed as explained previously in Section 2.3. The benchmarks used for the assessment of the performance are those in the 5D di-muon setup with $m_{\mu\mu} > 100\,\text{GeV}$ ($\mu\mu$-100). The same benchmark is also employed in Fig. 5 (bottom) for the comparison with the C2ST method. In the first row of Fig. 6, the baseline NPLM performance is compared with that of the NPLM variant that employs test-training splitting. We see that test-training splitting is detrimental for the sensitivity in all cases. The in-sample evaluation of the test statistics is thus found to play a key role for the sensitivity of the NPLM method. This finding explains the NPLM advantages in comparison with C2ST, but only partially. Indeed, the comparison with Fig. 5 reveals that the performances of the NPLM variant with test-train splitting are still superior to those attained by the C2ST method.

In the second row of Fig. 6 we display results (labelled as NPLM-half) obtained as follows: after training on the $(\mathcal{D}_{tr}, \mathcal{R})$ data sets we evaluate the test statistics on the same $(\mathcal{D}_{tr}, \mathcal{R})$ and we ignore the test data $\mathcal{D}_{te}$. This is of course equivalent to running the baseline NPLM method in a setup where only half of the data statistics is available, entailing sensitivity loss. The sensitivities obtained in this configuration are comparable to those of the NPLM method variant with test-train splitting. Splitting the data in equal portion for train and test is thus as ineffective as ignoring half of the available data points.

Inspecting the distribution of the test statistics in the different configurations, displayed in the three panels of Fig. 7, helps gaining an intuitive understanding of this result. The NPLM model learns from the training data and thus it is exposed to the statistical fluctuations that are present in the specific instance of the sample $\mathcal{D}$ used for training. When the data are R-distributed (light histograms), statistical fluctuations are in fact all there is to be learned, since $\mathcal{D}$ is distributed like the reference by construction. This produces a positive value for the test statistics evaluated on the same samples used for training (left and middle panels). When the data follow instead a different distribution than R, their statistical fluctuations are accompanied by a systematic discrepancy from the reference data. The test statistics (dark histogram) thus emerges as the sum of a contribution from statistical fluctuations that is typically as large in size as it was for R-distributed data, plus the systematic contribution. This latter contribution is responsible for the shift of the test statistics distribution, and in turn for the sensitivity of the method to the alternative data distribution. The amount of shift is controlled by the

data statistics and it is obviously larger when the full data statistics is employed (left panel) than if only half of the data (middle panel) are used. The continuous line is a $\chi^2$ distribution with a number of degrees of freedom given by the number of trainable parameters (weights and biases) of this particular neural network, a fully connected network with with five input nodes, three hidden layers with five nodes each and one output node ($3 \times 5 \times 6 + 6 = 96$).

When evaluated on test samples that are different from those used for training, the $t_{\text{LR}}$ distribution behaves instead as on the third panel of Fig. 7. The statistical fluctuations learned on the training data are different from the ones that are present in the test data. The trained model thus offers a poor description of the test data and $t_{\text{LR}}$ is typically negative, and especially so for R-distributed data because statistical fluctuations are all what the model learns. The systematic shift is still present in $t_{\text{LR}}$ when the data are distributed according to the alternative, because the systematic component of the data departures from the reference is present both in the training and in the test data. The size of this shift is again controlled by the statistics of the data, and of course the relevant data are only those used for evaluating the test statistics. These are half of the total in the test-train splitting setup, like in the configuration where only half of the available data are used. Since the shift controls the sensitivity, it is reasonable to expect, as we find, that the performances of the two configurations are comparable, and inferior to those of the baseline NPLM because of the effective loss of data.

## 3.2 Alternative test statistic

The ratio between the likelihoods is a standard measure of the relative fit quality of two hypotheses. Hence, the NPLM choice of the likelihood ratio test statistics $t_{\text{LR}}$ in Eq. (6) is very natural or even obliged from the perspective of a Neyman–Pearson test. From the viewpoint of classification instead, this choice is highly unnatural as emphasised in Section 2.1. We evaluate its impact on the performance by studying a number of NPLM variants that employ different test statistics to be evaluated on the trained classifier. We will consider variables that are either standard metrics for classification performance, or standard test statistics for one-dimensional GoF methods as in Friedman's proposal [3]. These variables and their performance are discussed in the next two sections in turn. In all cases, we proceed like in the baseline NPLM method with in-sample evaluation of the test statistics and the default selection of hyper-parameters for a direct assessment of the effect of the choice of the test statistics on the performance.

### 3.2.1 Standard classification metrics

The most standard classification metric is probably the classification accuracy. The detailed definition of this quantity must take into account that in our framework the $\mathcal{D}$ and $\mathcal{R}$ data sets are unbalanced and that the $\mathcal{D}$ size $N_{\mathcal{D}}$ is a discriminating variable. As in Sections 2.2 and 2.3, this leads to the definition

$$\text{ACC}_{\text{t}} = \frac{1}{N(R) + N_{\mathcal{D}}} \left[ \frac{N(R)}{N_{\mathcal{R}}} \sum_{x \in \mathcal{R}} \mathbb{I}[c_{\widehat{\mathbf{w}}}(x) < t] + \sum_{x \in \mathcal{D}} \mathbb{I}[c_{\widehat{\mathbf{w}}}(x) > t] \right], \qquad (14)$$

where $c_{\widehat{\mathbf{w}}}(x)$ is related to the trained NPLM model $f_{\widehat{\mathbf{w}}}(x)$ by Eq. (8). We consider two versions of the accuracy test statistics

$$t_{\text{ACC}}^{1/2} = \text{ACC}_{1/2}, \qquad t_{\text{ACC}}^{\max} = \max_{t \in [0,1]} [\text{ACC}_{\text{t}}]. \qquad (15)$$

The first one, with the threshold $t = 1/2$, is effectively equal to the C2ST test statistic in Eq. (12). The different normalisation accounts for the fact that the accuracy is evaluated on

the full data sets. The second tests statistics, $t_{\text{ACC}}^{\text{max}}$, employs a variable threshold optimised on the data under consideration.

Another widely employed metric for classification is the Area Under the ROC Curve (AUC). This the integral of the Receiver Operating Characteristic (ROC) curve, built by varying the classification threshold in the plane formed by the probability of false and of true positive. A straightforward adaptation of the standard AUC to our setup, where the samples are unbalanced and $N_{\mathcal{D}}$ is discriminant, is readily obtained as follows.

The false positive probability can be estimated as usual by the false positive rate

$$\text{FP}_{\text{t}} = \frac{1}{N_{\mathcal{R}}} \sum_{x \in \mathcal{R}} \mathbb{I}[c_{\widehat{\mathbf{w}}}(x) > \text{t}]. \tag{16}$$

The true positive probability is instead most conveniently expressed, for our purposes, in terms of the classification accuracy

$$\text{TP}_{\text{t}} = 2\text{ACC}_{\text{t}} + \text{FP}_{\text{t}} - 1. \tag{17}$$

The above equation relies on the regular definition of the accuracy as the average between the true positive and the true negative probabilities.

Eq. (14) provides a definition of the accuracy that accounts for the unbalance in the samples and retains the discriminating power of $N_{\mathcal{D}}$. We exploit the same definition in the calculation of the AUC as the integral of Eq. (17), obtaining the AUC test statistics

$$t_{\text{AUC}} = \int d\text{t} \left| \frac{d\text{FP}_{\text{t}}}{d\text{t}} \right| \text{TP}_{\text{t}} = 2 \int d\text{t} \left| \frac{d\text{FR}_{\text{t}}}{d\text{t}} \right| \text{ACC}_{\text{t}} - \frac{1}{2}. \tag{18}$$

Fig.s 11 and 12 displays the performances of the $t_{\text{ACC}}^{1/2}$, $t_{\text{ACC}}^{\text{max}}$ and $t_{\text{AUC}}$ test statistics on the benchmarks defined in Appendix A. The baseline NPLM performances are superior in general, and the classification metrics are exposed to dramatic failures in certain configurations. In particular, the classification metrics fail when the anomaly in the data emerges from a small fraction of signal events. We find here the same behaviour we observed in Section 2.3 for the C2ST method. We attributed this failure to the fact that individual data points weight the same in the classification accuracy metrics while points that are recognised as highly anomalous weight more in the likelihood ratio test statistics employed in the baseline NPLM method. The only exception to this trend is observed in the bottom panel of Fig. 11, showing the performances over the DQM benchmark. In this case, the performances of the classification metrics exceed those of the likelihood ratio test. We don't have a complete understanding of this result; a possible reason could be the significantly lower size of the data sample, which prevents an accurate approximation of the likelihood ratio shape. Further studies will be needed to understand this result.

### 3.2.2 One dimensional GoF variables

We now turn to a series of test statistics that are commonly employed for GoF on univariate data. Some of these variables need a slight adaptation to our setup, in which the GoF is addressed as a two-sample test—i.e., the reference distribution is not known in closed form but only a reference sample is available—and the total number of events in one of the two classes is a discriminating variable. These straightforward adjustments, and the definition of the test statistics, are reported in Appendix B. The univariate data sets $\mathfrak{D}$ and $\mathfrak{R}$ employed in the test statistics evaluation consist of the trained classifier $c_{\widehat{\mathbf{w}}}(x)$ evaluated on $\mathcal{D}$ and $\mathcal{R}$ as in Friedman's proposal [3].

The results are reported in Fig.s 13, 14 and 15, and described below.

$\chi^2$ **tests**    Histogramming the data $\mathfrak{D}$ and comparing the numbers of points in each bin with the reference hypothesis predictions is the most common GoF method. If the reference distribution is not known in closed form, the predictions can be readily obtained from the histogram of the $\mathfrak{R}$ data. The $\chi^2$ (see Eq. (B.1)) is the appropriate test statistic to perform this comparison. The method is subject to strong ambiguities in the choice of the binning strategy and in the number of bins $n_{bins}$. The standard binning strategy we employ is to adapt the width of each bin in order to make it contain exactly $N(R)/n_{bins}$ expected events in the R hypothesis. Different choices are considered for $n_{bins}$.

The results, presented in Fig. 13, display a considerable dependence of the sensitivity on the number of bins. The most effective choice of $n_{bins}$ depends on the source of the anomaly in the data, and therefore it can not be optimised because the essence of the GoF problem is to identify anomalous data sets without prior bias on the source of the anomaly. The observed sensitivity pattern follows the basic intuition: signals that are more localised require narrow binning to be seen, while distortions of the reference distribution that are spread on a wider range benefit from broader bins. The baseline NPLM results are always better or comparable to the $\chi^2$ test with the highest power curve in each benchmark problem.

**EDF tests**    Several univariate GoF methods are based on the Empirical cumulative Distribution Function (EDF). The most known example is the Kolmogorov–Smirnov (KS) test, others are the Cramer–von-Mises (CvM) and the Anderson–Darling (AD) tests. The performances of the corresponding test statistics (see eqs. (B.4,B.5)) are reported in Fig. 14.

EDF test statistics perform in general slightly better than $\chi^2$. In some cases they even display comparable performances to baseline NPLM, however none of the tests in this family is universally better than the others in all benchmarks. The different weights that the different tests give to the tails of the data distribution makes them more or less sensitive to different types of anomalies.

**Spacing statistics**    We finally consider the Moran (M) and the Recursive Product of Spacing (RPS) tests (eqs. (B.8,B.11)), as representative of and approach to GoF that exploits the distance between data points. We see in Fig. 15 that the performances are generically poor.

The survey performed in the present section enables us to reach a sharp conclusion on the crucial role played by the choice of the likelihood ratio test statistics (6) for the NPLM method performances. This choice, as well as the in-sample evaluation of the test statistics studied in the previous section, stems from the origin of NPLM as a Neyman–Pearson test rather than a classifier-based GoF method. Our results thus point to the superiority of Neyman–Pearson test approach.

## 4 Conclusions

We presented an initial assessment of the performances of the NPLM method for GoF, which is based on Neyman–Pearson testing, in comparison with the performances of classifier-based methods. The study has been carried out in two ways. First, through a direct comparison with the C2ST method, taken as representative of the classifier-based approach in its most direct and simple form. Second, by evaluating the impact of the most peculiar methodological choices of NPLM, which directly stem from its origin as a Neyman–Pearson test, on its performance. In particular, we analysed the effect of the in-sample evaluation of the test statistic and the usage of the likelihood ratio test statistics. Both, as foreseen in baseline NPLM, have been found to bring strong advantages. The NPLM method has been also compared, in Appendix C,

with several standard GoF methods that exist for one-dimensional variables. Overall better performances are obtained.

These results indicate that performing a Neyman–Pearson test of the reference hypothesis against a suitably designed set of alternative hypotheses, like NPLM does, is a powerful approach to GoF that deserves further studies.

It is not straightforward to extrapolate our findings beyond the specific framework we employed for the comparison and to turn them into general statements on the NPLM advantages in comparison with other approaches to GoF. An inherent feature of the GoF problem is the difficulty of identifying a sharp figure of merit to rank the effectiveness of different methods. The sensitivity to anomalous data drawn according to some specific alternative to the reference hypothesis is a clear figure of merit for hypothesis tests, because the goal of a hypothesis test is precisely to tell the reference hypothesis from one specific and pre-specified alternative. A GoF test aims instead at identifying data that are anomalous with respect to the reference hypothesis, regardless (if possible) of the specific alternative distribution according to which the data are truly distributed. The sensitivity to anomalous data is still the figure of merit. However, a wide set of different alternative hypotheses for their true distribution must be considered for a meaningful assessment. Still, the assessment will never be fully conclusive as the result does depend in general on the alternatives that have been considered. NPLM has been found generally better than the other methods over all the benchmark problems we studied. Furthermore, none of the other methods performs comparably well as NPLM on all the benchmarks. Nevertheless, these results could in principle change if the comparison was extended to include other benchmarks.

One bias that certainly affects our selection of the benchmarks is the origin of NPLM as a method to search for new physics in high energy collider data. Many of our benchmarks have thus been selected, in the previous NPLM literature, to be representative of this type of problems. However, we also considered the DQM benchmarks, which are unrelated with new physics at colliders and are based on natural data collected at a muon detector. Furthermore, our benchmark problems do not seem to display any strong peculiarity, apart from being arguably difficult GoF problems: the reference and the alternative distributions are on the same support and the discrepancies emerge either from a small localised excess (or deficit) of events or from a distributed distortion of the reference distribution. There is no reason to expect that our findings would be radically different on different problems as our benchmarks are not peculiar and are reasonably varied. However, it should be noted that we only deal with data of limited dimensionality. NPLM has been applied to data with up to 100 dimensions so far, and a comparison with other GoF methods suited to deal with this dimensionality could be considered in the future. Modifications to the NPLM methodology to handle extremely high dimensional problems, such as the validation of image generation models, will be considered in future works.

Previous works [12–16] remarked a certain degree of uniformity of the NPLM response to qualitatively different types of anomalies injected on top of the same reference distribution. This uniformity is defined—see Appendix A for a review—as a correlation between the median sensitivity of NPLM and the median sensitivity, $Z_{id}$, of the fully optimal hypothesis test designed to detect the specific source of anomaly under consideration. We do not observe this uniformity for C2ST nor for any of the other GoF methods we studied in this paper as NPLM variants.[6] The complete loss of sensitivity of certain methods to specific type of anomalies, accompanied with a more moderate degradation of their sensitivity in comparison with baseline NPLM, exemplifies this lack of uniformity. We currently lack the theoretical understanding needed to

---

[6]The train-test split configuration studied in Section 3.1 is an exception. This is because the train-test split configuration performances are understandably comparable with those of NPLM with half statistics, which are inferior to baseline NPLM but still universal.

accompany the empirical evidence gathered in this work about this phenomenon with a more quantitative study. The proper quantification of $Z_{id}$ is often in itself a technical challenge. Advances in this speculative direction are potentially very interesting and should be pursued.

## Acknowledgments

We thank Bob Cousins and Louis Lyons for useful discussions and suggestions.

**Funding information** A.W. is supported by the grant PID2020-115845GB-I00/AEI/ 10.13039/501100011033. G.G. and M.P. are supported by the European Research Council (ERC) under the European Union's Horizon 2020 research and innovation program (grant agreement no 772369). M.L. acknowledges the financial support of the European Research Council (grant SLING 819789).

## A Benchmarks

The only way to assess the performances of GoF methods is to study their ability to identify anomalous data drawn from a distribution that is different from the reference one. Benchmark problems and the corresponding benchmark data sets for GoF performance studies are thus defined by one given R hypothesis for the reference distribution, plus one given alternative hypothesis.

The selection of the benchmarks is unavoidably arbitrary. Therefore, it must be reasonably justified considering that the purpose of GoF is to spot out generic data departures from the R hypothesis regardless of the specific alternative that underlies their distribution. We thus organise our benchmark problems as specific alternatives that could be possibly encountered in a number of GoF setups. One setup is defined by one chosen reference probability distribution and number of expected events N(R). Additionally, since the GoF methods we investigate are in fact two-sample tests, the size of the available $\mathcal{R}$ sample—or the ratio $N_{\mathcal{R}}/N(R)$—is also specified in the definition of the setup.

The GoF setups and benchmarks employed in this paper are described in the following sections. It should be noted that the vast majority of these benchmarks were defined and studied in previous NPLM works [12–16], and they have not been designed specifically for the comparative studies of the present article.

Within each setup, we would like our GoF method to be sensitive to data generated according to any possible alternative distribution. Clearly in practice we can only study the sensitivity to a limited number of different alternatives, to be selected with care trying to cover qualitatively different types of data anomalies. For instance, anomalies emerging from small discrepancies of the distribution over a wide region of the $x$ variable are different from sharply localised signals emerging in the tail of the $x$ distribution, or in the bulk. A GoF test can be more or less sensitive to anomalies in one class than in the other. This is confirmed by the results of the present article: the sensitivity degradation of the many methods we studied—relative to the baseline NPLM sensitivity—is more or less pronounced for the different alternatives we considered in each setup. This result also confirms, a posteriori, the validity of our benchmark problems selection as probing different sources of anomalies in the data that can be easier or harder to see for a generic GoF method.

A different question is which level of sensitivity is legitimate to expect from a "good" GoF method for each of the different alternatives that we might consider in one given setup. One should keep in mind that deviations from the reference distribution can be arbitrarily small

and not detectable from finite samples. Therefore, some sensitivity can be expected for alternatives that are easy enough to be detected, according to some notion of "easiness". Such a notion can be naturally introduced for comparing alternative distributions of a specific type, based on the value of their adjustable parameters. For instance, sharp peaks are characterised by the area below the peak region, and higher peaks are obviously easier to see. On the other hand, a different and more abstract notion is needed in order to compare alternative hypotheses in different classes, which are controlled by different and not commensurable adjustable parameters.

In previous works on NPLM [12–16] we introduced and employed the notion of ideal sensitivity or ideal median $Z$-score, $Z_{id}$. The definition of this quantity emerges from the powerful result known as the Neyman–Pearson lemma [9], identifying the hypothesis test strategy that is the most sensitive, defined as the most powerful at any given test size. The sensitivity of such optimal test, $Z_{id}$, serves two purposes. On the one hand, it can be used for an objective evaluation since it is an upper bound to any GoF method. However, it also allows to compare different alternative hypotheses by establishing how hard it is to detect them. Therefore, a "good" GoF method should respond uniformly—i.e., with comparable values of $Z$—to all the alternatives with comparable $Z_{id}$. This uniform response is qualitatively observed for the NPLM method within each of the GoF setups we studied (but not across different setups). This is reviewed in the next sections.

## A.1 Expo

This is a simple univariate setup that represents an energy or transverse-momentum spectrum that falls exponentially. Such types of distributions are fairly common in collider physics experiments. Studying GoF techniques in this setup is thus illustrative of some of the challenges associated with the search for new physics at these experiments. NPLM performances in the Expo setup were studied in Refs. [12, 14]. A minor extension of these studies has been performed for this paper as described below.

The reference distribution is

$$n(x|R) = N(R)e^{-x}, \tag{A.1}$$

where the expected number of collected events in the reference hypothesis, N(R), is set at 2000. The $\mathcal{R}$ sample is composed of $N_{\mathcal{R}} = 100\,N(R)$ events. We consider (see Fig. 8) a total of five alternative hypotheses for the true data distribution, mimicking qualitatively different ways in which the data can depart from the reference hypothesis expectation. We refer to these deviations as *signals* and they are modelled as follows:

**H$_1$**: a peak in the tail. This is modelled by adding to the exponential reference distribution a Gaussian distribution with mean 6.4, standard deviation 0.16 and N(S) = 10 expected number of events. The total number of expected events in the H$_1$ hypothesis is N(H$_1$) = N(R) + N(S) = 2010.

**H$_2$**: a quadratically growing excess in the tail, modelled as an additive contribution to the reference distribution that is proportional to $x^2 e^{-x}$ and normalised to N(S) = 90 signal events. The total number of events is N(H$_2$) = N(R) + N(S) = 2090.

**H$_2'$**: similar to H$_2$ but with only shape effects. This is engineered by lowering the area of the exponential component of the distribution to N(R)$'$ = 1890 and adding N(S) = 110 events with the $x^2 e^{-x}$ distribution. In this way, N(H$_2'$) = N(R) and the total number of events is not a discriminating variable. Notice that, unlike H$_1$ and H$_2$, the H$_2'$ distribution is not an additive modification of the reference distribution.

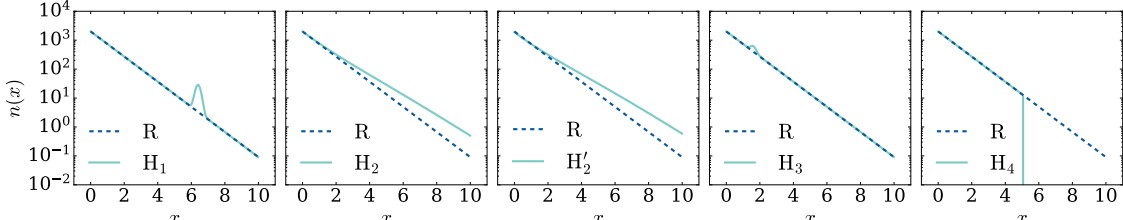

Figure 8: The five alternative distributions that are considered in the Expo setup. The reference distribution is shown dashed in all plots.

**H$_3$**: a peak in the bulk of the reference distribution, modelled as a Gaussian with mean 1.6 and standard deviation 0.16. The expected number of signal events is N(S) = 90.

**H$_4$**: a defect in the tail of the distribution obtained by cutting the reference exponential distribution above the threshold $x > 5.07$. The total number of expected events is reduced, relative to N(R), to N(H$_4$) = 1987.

The specific tuning of the parameters—such as the number of expected signal events, or the value of the threshold in the case of H$_4$—that define these alternative hypothesis is such that the median ideal $Z$-score of all the alternatives is around $5\sigma$. The ideal $Z$-score $Z_{\mathrm{id}}$ is obtained as a direct application of the Neyman–Pearson lemma, exploiting the analytic knowledge of the reference and of the alternative distributions. Namely, we perform a hypothesis test using the test statistic [9]

$$t_{\mathrm{H}} = 2\log\left[\frac{e^{-\mathrm{N(H)}}}{e^{-\mathrm{N(R)}}}\prod_{x\in\mathcal{D}}\frac{n(x|\mathrm{H})}{n(x|\mathrm{R})}\right]. \tag{A.2}$$

For each H = H$_{1,...,4}$, we compute empirically the distribution of the corresponding variable $t_{\mathrm{H}}$ under the R hypothesis by toy experiments. We then compute the median p-value on data that follow the alternative distribution, and eventually the median $Z$-score defined as in Eq. (13). The results are reported in the second column of Table 1.

The second column of the table displays the median $Z$-scores of the NPLM tests when the data are distributed according to the alternative hypotheses. These results correspond to the point on the $Z_\alpha$ axis where the power curves cross 1/2. They are obtained with a $(1,4,1)$ network and weight clipping regularisation parameter of 8. The hyperparameters selection has been performed with the standard NPLM criterion of $\chi^2$-compatibility [13,14]. No preprocessing of the variables given as input to the network is performed since the data naturally have unit mean and unit variance. Data are instead shifted and scaled to have zero mean and unit variance—for R-distributed data—in the other NPLM applications considered in this paper.

We also employ the kernel-based NPLM implementation in the Expo setup. This is characterised by three main hyper-parameters that are set to the values $(M, \sigma, \lambda) = (5000, 2.3, 10^{-10})$, which we obtain following the general hyper-parameters selection prescription detailed in Refs. [15,16].[7]

---

[7]The exact values are not the ones from the original study in Ref. [15], due to the different range of the input variable.

Table 1: Z-scores for the Expo setup.

| Alternative | $Z_{\mathrm{id}}$ | $Z$ | $Z/Z_{\mathrm{id}}$ |
|:---:|:---:|:---:|:---:|
| **Expo** | | | |
| $H_1$ | 4.7 | 2.5 | 0.5 |
| $H_2$ | 4.4 | 2.5 | 0.6 |
| $H_2'$ | 4.4 | 3.0 | 0.7 |
| $H_3$ | 5.2 | 3.9 | 0.8 |
| $H_4$ | 4.5 | 2.6 | 0.6 |

## A.2 $\mu\mu$

This is a more complex setup inspired by the realistic problem of a model-agnostic search for physics Beyond the Standard Model (BSM) with LHC data. In particular, the setup targets LHC final states with two muons of opposite charge. The goal is to assess whether the distribution of the kinematical variables that characterise the muons are well described by the corresponding Standard Model (SM) predictions. This use case was proposed in [13] and further investigated in Refs. [14, 15].

The data sets are available on Zenodo [19] and consists of collections of the five variables describing the kinematics of two muons produced in the final states of proton-proton collisions at the LHC: the transverse momenta of the two muons ($p_{T,1}$, $p_{T,2}$), their pseudorapidities ($\eta_1$, $\eta_2$) and their relative azimuthal angle $\Delta\phi_{1,2}$.

The reference hypothesis is the adequacy of the SM as a description of the process. Therefore the reference-distributed data are obtained by Monte Carlo simulations that are based on the SM theory. The main SM process that contributes to this final state is the Drell–Yan process where the muons emerge from the decay of a virtual photon or $Z$-boson particle generated in the annihilation of a quark-antiquark pair in the colliding protons. The SM theory also predicts the total number of events with two muons that are expected to be observed after a certain number of protons have been collided. More precisely, the number of expected events is proportional to the *integrated luminosity* of the LHC collider that is employed in the analysis. By varying the data luminosity that we decide to employ, we can thus control the total number of expected events N(R) of our GoF case study.

Additionally, we can decide to restrict our analysis to the subset of the events that satisfy certain cuts on the kinematics of the muons.[8] Two choices are considered, defining two distinct GoF setups denoted as $\mu\mu$-60 and $\mu\mu$-100. In the first case, only events corresponding to an invariant mass of the two muons larger than 60 GeV are kept. This selection rule was already considered in [13], and it corresponds to the threshold below which muons are hard to identify experimentally and the SM predictions for their distribution become more difficult to obtain. This cut does not exclude the mass of the $Z$-boson particle at approximately 90 GeV, which can thus be produced resonantly and dominate the composition of the $\mu\mu$-60 reference distribution. In the second case, we only retain events with an invariant mass of the two muons above 100 GeV. This cut was introduced in [14] and excludes the resonant $Z$-boson production entailing a strong change in the reference distribution compared to the other case. In order to maintain a comparable number of expected events in the two setups, the integrated luminosities have been chosen to be 0.35 fb$^{-1}$ for the $\mu\mu$-60 and 3.5 fb$^{-1}$ for the $\mu\mu$-100 setup. This choice corresponds to N(R) = 18740 and N(R) = 84530 in the two setups.

The alternative hypotheses we would like to be sensitive include phenomena not predicted by the SM, but by an arbitrary extension of the latter (a BSM model). Simple BSM benchmark

---

[8]Acceptance cuts $p_T > 20$ GeV and $|\eta| < 2.4$ are always applied.

points are thus employed to generate anomalous data sets. Namely, we consider a so-called $Z'$ model where a new massive spin-one particle is present with the same couplings of the SM $Z$-boson. A $Z'$ particle with 200 GeV or 300 GeV mass is considered in order to probe its effect in different regions of the reference SM distribution. We also consider a non-resonant deformation of the SM distribution due to a new EFT interaction operator of dimension-6. Specifically, we consider an operator $c_W / \Lambda^2 J_{L\mu}^a J_{La}^\mu$ where $J_{La}^\mu$ is the $SU(2)_L$ SM current. The energy scale $\Lambda$ is fixed at 1 TeV and the Wilson coefficient $c_W$ determines the strength of the effect. In the $\mu\mu$-60 and $\mu\mu$-100 setups, the following benchmarks are considered:

**$\mu\mu$-60:**

**$Z'$-2-a**:
a $Z'$ with 200 GeV mass and a cross-section such as to produce N(S) = 80 expected signal events in addition to the N(R) = 18740 events from the SM background.

**$Z'$-2-b**:
a $Z'$ with 200 GeV mass and N(S) = 160.

**$Z'$-3**:
a $Z'$ with 300 GeV mass and N(S) = 40.

**EFT-a**:
the EFT operator with Wilson coefficient $c_W = 1.2$. The total number of expected events in the presence of the new operator is 18747.

**EFT-b**:
the EFT operator with Wilson coefficient $c_W = 1.5$. The total number of expected events becomes 18783.

**$\mu\mu$-100:**

**$Z'$-2-a**:
a $Z'$ with 200 GeV mass and N(S) = 120 expected signal events. In the $\mu\mu$-100 setup, the number of expected SM background events is N(R) = 84530.

**$Z'$-2-b**:
a $Z'$ with 200 GeV mass and N(S) = 240.

**$Z'$-3-a**:
a $Z'$ with 300 GeV mass and N(S) = 60.

**$Z'$-3-b**:
a $Z'$ with 300 GeV mass and N(S) = 120.

**EFT**:
the EFT operator with Wilson coefficient $c_W = 1.5$. The total number of expected events becomes 87290.

All benchmarks considered in $\mu\mu$-60, except for **$Z'$-2-b**, were previously studied in [13], while the benchmarks in $\mu\mu$-100 have been defined so that the ideal reaches are comparable to the $\mu\mu$-60 ones.

As in the Expo setup, we need to quantify the ideal $Z$-score of our benchmarks. However, this is more difficult to achieve in this case because the distributions of the reference and of the alternative hypotheses are not available in closed form. We thus have to rely on approximations of the ideal $Z$-scores, obtained as follows. For the $Z'$ case, we assume based on experience

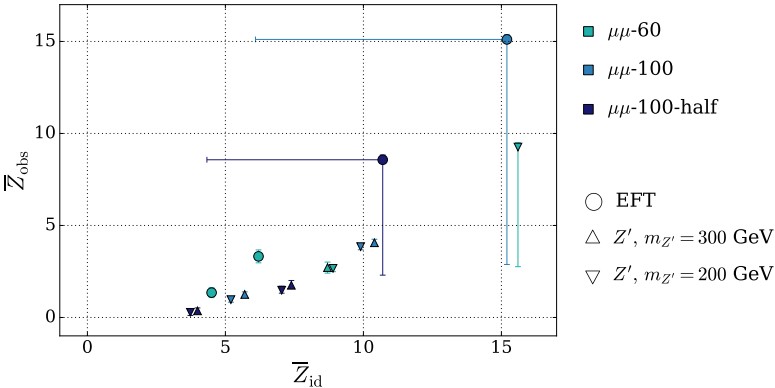

Figure 9: Correlation between the median ideal sensitivity and the median NPLM-NN sensitivity in the $\mu\mu$-60, $\mu\mu$-100 and $\mu\mu$-100-half setups.

that the invariant mass of the muons is the most effective variable to identify the presence of the $Z'$ in the data, and that a simple cut-and-count strategy in an invariant-mass window around the $Z'$ mass offers nearly optimal sensitivity. The width of the window is optimised for each benchmark signal. The sensitivity we obtain by counting events in the window and comparing with the reference model prediction for the expected values is reported in Fig. 9. Notice that very high $Z_{\text{id}}$ sensitivities can be estimated with the cut-and-count strategy thanks to the analytic knowledge of the (Poisson) distribution of the number of events in the window.

The situation is different for the EFT benchmarks. Also in this case, it could be reasonable— though more questionable—to assume that the invariant mass is the only discriminant variable, but a simple cut-and-count strategy would not produce nearly optimal sensitivity. We need to bin the invariant mass and compare the counts in each bin with the reference predictions. The number of bins and the binning strategy can be optimised. However, the distribution of the test statistic to be employed for the comparison is not known in closed form. The sensitivity must therefore be estimated empirically by running toy experiments to determine the distribution of the test statistic in the R hypothesis. This prevents in practice to estimate the sensitivity if it is approximately above $5\sigma$. When this occurs—specifically, for the EFT benchmark in the $\mu\mu$-100 setups— we report in Fig. 9 a very rough estimate of the sensitivity based on fitting the empirical distribution of the test statistic with a Gaussian distribution. With this approach, we can go beyond what is allowed by the empirical estimate (EFT data points for $\mu\mu$-100 and $\mu\mu$-100-half on the right half of the Figure). In these cases, the lower bound on the sensitivity that is obtained empirically is indicated by the lower edge of the error bars. Overall, our $Z_{\text{id}}$ estimates are not accurate enough to draw robust conclusion on the uniformity of the NPLM sensitivity in this setup. Nevertheless, the correlation between $Z_{\text{id}}$ and the median NPLM sensitivity is reported in Fig. 9.

The NPLM-NN hyper-parameters selection for this setup was discussed in [13] (see also [14]): we use a fully connected feedforward neural network with three layers of five nodes each and weight clipping is set to 2.15.

The $\mu\mu$ goodness of fit setups are employed extensively in the main text to illustrate the NPLM sensitivity. In Section 3.1 (see in particular the bottom plots in Fig. 6) we also use NPLM to address the $\mu\mu$-100 GoF problem using only half of the data statistics. This is effectively a different setup, denoted as $\mu\mu$-100-half, where N(R) and the number of expected events in all the alternative hypotheses is reduced by a factor 2. In the $\mu\mu$-100-half setup, the weight clipping regularisation parameter resulting from $\chi^2$-compatibility is equal to 2.4.

The hyper-parameters of the kernel-based implementation NPLM-KM in all the three $\mu\mu$ setups are set to $(M, \sigma, \lambda) = (2 \times 10^4, 3, 10^{-6})$ like in Ref. [15].

### A.3 DQM

Goodness of fit methods can be used to monitor the quality of the data produced by a complex apparatus such as a particle detector, i.e. for Data Quality Monitoring (DQM). The R hypothesis is that the detector is operating in normal design conditions, therefore the R-distributed samplings of the detector readout $x$ are obtained from the normal apparatus operations. Anomalies in the $x$ distribution can emerge from many different sources of technical failures.

The DQM GoF setup was introduced in Ref. [16] to exemplify this use case. It is based on natural data, available on Zenodo [20], from cosmic-ray muons at a drift tube chamber. The five variables collected by the apparatus are four drift times associated with the muon track and the angle formed by the muon track with the vertical axis. The data set consists of $3 \times 10^5$ data points collected under reference working conditions, and smaller (order $10^4$ points) data sets where some of the detector parameters are artificially altered. In this work we consider two types of anomalous working conditions: the cathod (Ca) anomaly where the voltage of the cathodic strips is reduce to 75% of their nominal value and the threshold (Thr) anomaly where the frontend threshold (is lowered to 75% of its nominal value. More anomalies have been studied in Ref. [16].

Using these data, the DQM setup for GoF is defined as follows. The data set size is fixed to $N_{\mathcal{D}} = 250$, corresponding to one possible choice of the size of the data batches to be monitored. Unlike in previous setups, $N_{\mathcal{D}}$ is fixed and is not a random variable. The size of the $\mathcal{R}$ data set is set to $N_{\mathcal{R}} = 2000$. A total of four benchmark anomalous data distributions are considered:

**Ca-50:** a mixture of 50% of Ca anomalous data and 50% of R-distributed data.

**Ca-70**: a mixture of 70% of Ca anomalous data and 30% of R-distributed data.

**Thr-50**: a mixture of 50% of Thr anomalous data and 50% of R-distributed data.

**Thr-70**: a mixture of 70% of Thr anomalous data and 30% of R-distributed data.

We exclusively employ the NPLM implementation based on kernel method to address the DQM GoF problem. As in [16], the hyper-parameters are set to $(M, \sigma, \lambda) = (2000, 4.5, 10^{-6})$.

## B  One dimensional GoF methods

The literature on goodness of fit tests for a one-dimensional variable $x$ is vast. Studying all the existing tests would be impossible and not particularly useful. We instead select a number of tests that are representative of different approaches that have been considered by statisticians throughout the years. These are described in turn in the rest of this appendix.

It should be kept in mind that ours is not strictly a GoF problem, because the data distribution in the reference hypothesis is not known in closed form. It is rather a two-sample test—though with unbalanced samples—between two sets $\mathfrak{D}$ and $\mathfrak{R}$ of univariate data of size $N_{\mathcal{D}}$ and $N_{\mathcal{R}}$, respectively. For the studies in Section 3.2.2, these data are obtained by applying the trained classifier to the $\mathcal{D}$ and $\mathcal{R}$ sets. For those in Appendix C, the data are directly provided by the $\mathcal{D}$ and $\mathcal{R}$ sets in univariate GoF problem.

Furthermore, in our setup $N_{\mathcal{D}}$ is a Poisson-distributed variable with expected $N(R)$ in the reference hypothesis, and not a pre-specified fixed number. Some of the classical GoF strategies we consider require minor adjustments in order to deal with this peculiarity, and in order to be used for two-sample tests instead of GoF tests.

**$\chi^2$ tests** Histogramming the data $\mathfrak{D}$ and comparing the numbers of points in each bin with the reference hypothesis predictions is probably the most common GoF method in high-energy physics. The corresponding test statistics is the Pearson's $\chi^2$

$$t_{\chi^2} = \sum_{i=1}^{n_{\text{bins}}} \frac{(o_i - e_i(\text{R}))^2}{e_i(\text{R})}, \tag{B.1}$$

where the $o_i$ is the number of points in the $\mathfrak{D}$ set that fall in the $i$-th bin and $e_i(\text{R})$ is the expected number in the R hypothesis. If the reference distribution is not known in closed form, the predictions for the expected numbers $e_i(\text{R})$ are readily obtained from the histogram of the $\mathfrak{R}$ data. The $t_{\chi^2}$ test statistics is thus automatically suited to be employed for a 2ST. Furthermore, it is automatically sensitive to the total number of observed events $N_{\mathcal{D}}$ and to its possible departures from the expected total number in the R hypothesis, $N(\text{R})$.

The $\chi^2$ method is subject to strong ambiguities in the choice of the binning strategy and in the number of bins $n_{\text{bins}}$. We construct non-overlapping bins that cover the entire support of the variable $x$ in such a way that each of them contains exactly $e_i(\text{R}) = N(\text{R})/n_{\text{bins}}$ expected events in the R hypothesis. Namely, we compute the $i/n_{\text{bins}}$ percentiles of the $\mathfrak{R}$ data set, for $i = 1, \ldots, n_{\text{bins}} - 1$ and we use them as the upper extreme of the $i$-th bin and as the lower extreme of the $i-1$-th bin. The first and the last bin extend up to the lower and upper boundary of the support of $x$. The number of bins $n_{\text{bins}}$ is left as a free parameter of the test.

Under certain conditions, the $t_{\chi^2}$ distribution in the R hypothesis can be accurately estimated by asymptotic formulas. We do not employ these formulas and opt for a fully empirical determination of the distribution by toy experiments.

**EDF test** Several GoF tests, including Kolmogorov–Smirnov (KS), are based on the Empirical cumulative Distribution Function (EDF) of the data $\mathfrak{D}$. The EDF function is defined by counting the number of instances of $x$ in $\mathfrak{D}$ that fall below a threshold $y$. Specifically, we define

$$\text{EDF}_{\mathfrak{D}}(y) = \frac{1}{N(\text{R})} \sum_{x \in \mathfrak{D}} \mathbb{I}(x < y), \tag{B.2}$$

where $\mathbb{I}$ is the indicator function. Notice that this is different from the regular definition because of the $1/N(\text{R})$ normalisation factor, which would normally read $1/N_{\mathcal{D}}$.

The reason for this unconventional definition is that the regular normalisation washes out the sensitivity to the number of observed data points, which we instead want to retain. Stated differently, in our case we are not interested in the regular cumulative distribution function defined as the integral of the probability distribution function, but rather the integral of the number density distribution. Eq. (B.2) is in fact an approximation of the latter integral, times the fixed constant $1/N(\text{R})$.

The regular EDF-based GoF tests are based on the comparison of the EDF function (B.2) with the cumulative distribution function of the variable $x$ in the R hypothesis. This is not known in closed form in our case, but it can be estimated using the $\mathfrak{R}$ sample:

$$\text{CDF}_{\text{R}}(y) \approx \text{EDF}_{\mathfrak{R}}(y) = \frac{1}{N_{\mathcal{R}}} \sum_{x \in \mathfrak{R}} \mathbb{I}(x < y). \tag{B.3}$$

By employing the $\text{EDF}_{\mathfrak{R}}$ function in place of $\text{CDF}_{\text{R}}$, all the regular test statistics for EDF tests can be employed. We consider the following options:

**Kolmogorov–Smirnov (KS):** the test statistics is

$$t_{\text{KS}} = \max_x |\text{EDF}_{\mathcal{D}}(x) - \text{EDF}_{\mathfrak{R}}(x)|. \tag{B.4}$$

**Cramer-von-Mises (CvM) and Anderson-Darling (AD):** the test statistic takes the following generic form

$$t = N_{\mathcal{D}} \int w(x) \left[ \mathrm{EDF}_{\mathcal{D}}(x) - \mathrm{EDF}_{\mathfrak{R}}(x) \right]^2 \frac{d\mathrm{EDF}_{\mathfrak{R}}}{dx} dx \,. \tag{B.5}$$

The weight function $w(x)$ is equal to 1 in CvM, and equal to

$$w_{\mathrm{AD}}(x) = \frac{1}{\mathrm{EDF}_{\mathfrak{R}}(x)\left(1 - \mathrm{EDF}_{\mathfrak{R}}(x)\right)} \,, \tag{B.6}$$

in the case of the AD test.

**Spacing statistics**  The last family of tests we consider is based on spacing statistics. Here, the degree of agreement of the data sample $\mathfrak{D}$ with the reference distribution is quantified based on the distance between the events in the data sample. These tests are constructed by first mapping the data points into a space where they are uniformly distributed under the reference hypothesis. Departures from uniformity in the new space signals disagreement with the reference distribution. The map is defined by the cumulative in the R hypothesis, $\mathrm{CDF}_R(x)$. When this is not available, as in our case, one can obtain an approximate empirical map using the $\mathfrak{R}$ sample, namely

$$x \;\rightarrow\; u = \mathrm{EDF}_{\mathfrak{R}}(x)\,. \tag{B.7}$$

Operating with the transformation on the $\mathfrak{D}$ data set, and sorting the result such that $u_i < u_{i+1}$, we define a sample of spacings (i.e., distances) $S = \{s_i\}_{i=1}^{N_{\mathcal{D}}+1}$ with
$s_1 = u_1$, $s_{N_{\mathcal{D}}+1} = 1 - u_{N_{\mathcal{D}}}$, and $s_i = u_{i+1} - u_i$. Notice that the sum of all elements in $S$ is equal to one by definition. The elements of the sample $S$ are employed to define the test statistic. In the simplest version, the **Moran (M) test**, the test statistics is the negative sum of the logarithms of each spacing

$$t_{\mathrm{M}} = -\sum_{s \in S} \log(s)\,. \tag{B.8}$$

A more complex version is the **recursive product of spacing (RPS) test**, which was recently proposed in Ref. [21]. The RPS proposal is to include spacings of higher order—i.e., distance between non-consecutive points—in the test statistics, in a recursive manner. The recursion defines a total of $N_{\mathcal{D}} + 1$ samples of distances, $S^{(k)}$, where $S^{(0)} = S$. The $S^{(k)}$ sample contain $N_{\mathcal{D}} + 1 - k$ points

$$S^{(k)} = \left\{ s_i^{(k)} \right\}_{i=1}^{N_{\mathcal{D}}+1-k}\,, \tag{B.9}$$

defined by the recursion relation

$$s_i^{(k+1)} = \left( s_i^{(k)} + s_{i+1}^{(k)} \right) \Big/ \sum_{j=1}^{N_{\mathcal{D}}-k} \left( s_j^{(k)} + s_{j+1}^{(k)} \right)\,. \tag{B.10}$$

The normalisation ensures that the sum of all the elements of each $S^{(k)}$ set is equal to one. The RPS test statistics is given by sum of the Moran statistics for each set

$$t_{\mathrm{RPS}} = -\sum_{k=0}^{N_{\mathcal{D}}} \sum_{s \in S^{(k)}} \log(s)\,. \tag{B.11}$$

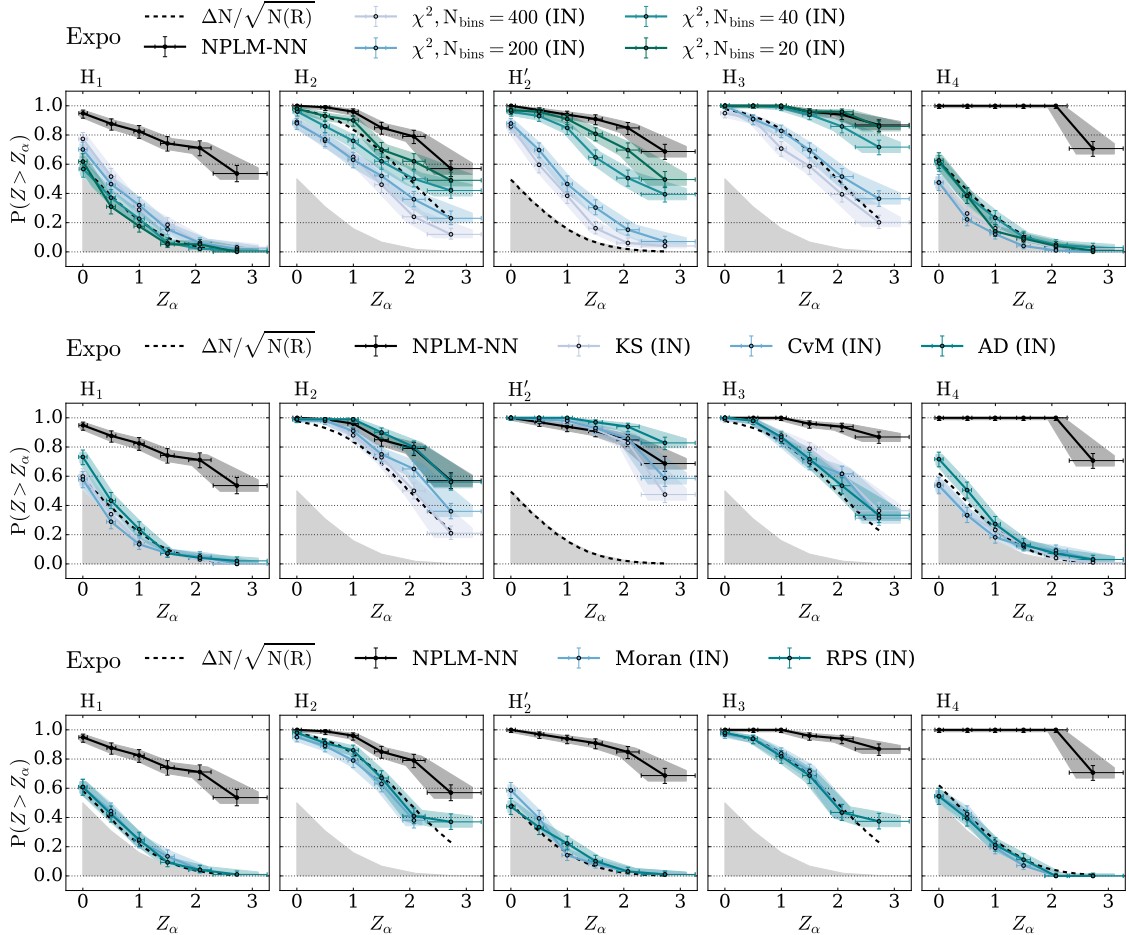

Figure 10: Power curves for the traditional GoF tests on the five signal benchmarks of the Expo setup, compared with NPLM. The tests are performed on the features given as inputs to NPLM (IN).

## C  Comparison with 1D GoF tests

When the data are one-dimensional, GoF is sometimes regarded as a solved problem. On one hand, it is arguably true that a careful inspection of a one-dimensional set of data would always allow the analyser to identify departures from the reference hypothesis expectations. Visualising histograms with different binnings, looking for outliers or anomalous concentrations of data points are common analysis strategies that do not even necessarily require a rigorously defined GoF test. On the other hand, human inspection of the data could be unfeasible in some cases and the process of anomalous data detection might need to be automated. Furthermore, a standardised quantification of the anomaly could be required, which human analysers can hardly provide. It is thus important to study and develop GoF methods also in the simpler setup of univariate data. This appendix is devoted to the study of the Expo setup defined in Section A.1. We already discussed the NPLM sensitivity in this setup, and the sensitivity of a number of NPLM variants inspired by the classifier-based approach. We discuss here how the Expo GoF problem is addressed by regular GoF methods—defined in Appendix B—directly applied to the data and without employing classifiers.

The results are reported in Fig. 10, and compared with the NPLM-NN results. We see that the NPLM power curves are always well above those of the spacing statistics tests. The $\chi^2$ tests can give comparable performances to NPLM for suitable (but selected a posteriori) $n_{bins}$, but

they are exposed to dramatic failures in particular in the $H_1$ and $H_4$ benchmarks. EDF-based tests perform better. In particular, the AD test is equivalent to NPLM for $H_1$ and $H_4$, slightly better than NPLM in the case of $H_2$ and $H_2'$, but it fails rather strongly on the $H_3$ benchmark. Overall, the results confirm the general pattern observed in the rest of the paper: the NPLM method performs well and it is much less exposed than other methods to strong sensitivity failures for specific types of anomalous data. It thus qualifies as a "good" approach to GoF.

# D  Figures

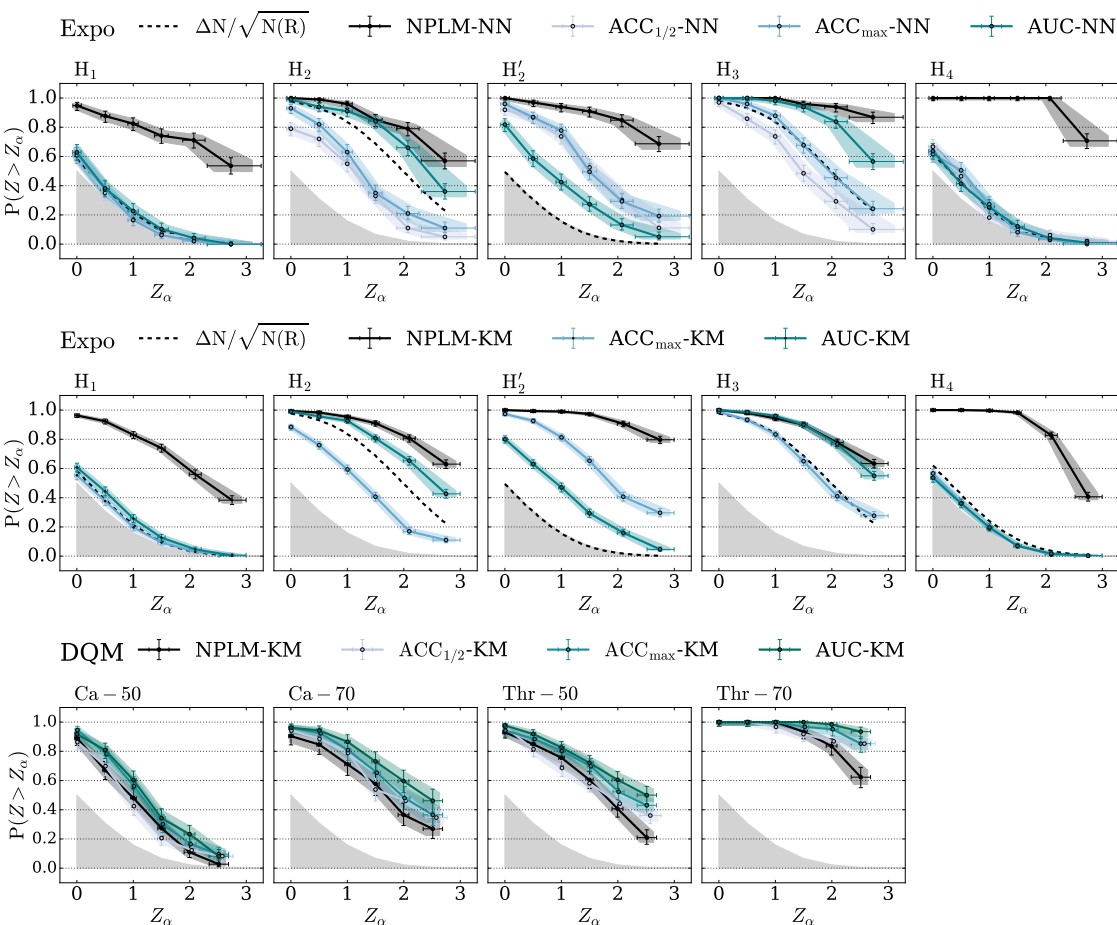

Figure 11: **Classification metrics:** Power curves for the NPLM variants that employ standard classification metrics computed over the output of the model (OUT) as test statistics, compared with baseline NPLM.

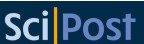

**Figure 12: Classification metrics:** Power curves for the NPLM variants that employ standard classification metrics as test statistics, compared with baseline NPLM.

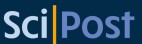

Figure 13: $\chi^2$ **metrics:** Power curves for the NPLM variants that employ the $\chi^2$ test statistics computed on the output of the model (OUT), compared with baseline NPLM.

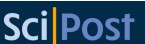

Figure 14: **EDF metrics:** Power curves for the NPLM variants that employ EDF-based test statistics, compared with baseline NPLM.



Figure 15: **Spacing test metrics:** Power curves for the NPLM variants that employ the spacing test statistics, compared with baseline NPLM.

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
