# Peer review of "Goodness of fit by Neyman-Pearson testing"

_SciPost Physics, doi:SciPost Phys. 16, 123 (2024)_

## Round 2 · Author Response

Dear Referees and Editor,

Thank you for for acknowledging the relevance of our work, the careful reading of the manuscript and the many comments that allowed us to increase its quality.

We report here some comments regarding specific points that deserve further clarifications.

--Referee 1 - Figure 7, EFT : The amount of signal injection is such that all models demonstrate a high power in this particular case. Note how this is not guaranteed, as shown for example in Figures 13 and 14 for different tests on the same data.

--Referee 1 - Paragraph 3.2: We thank the referee for the comments as they help us to clarify crucial aspects of our methodology. The correct procedure is the following: - We use a supervised ML model with a loss such that the ratio of the data-generating densities is approximated at the end of the training, i.e. f from eq. 1 (note that the actual likelihood ratio includes a Poisson distribution that models the fluctuations in the number of collected data points, as shown in eq. 4). The loss is a weighted logistic for the model based on kernel methods (eq. 2) and the “likelihood-ratio loss” for the one based on neural networks (eq. 3). - In both cases, we plug the learned f in the expression for the extended likelihood ratio (eq. 6) which can then be seen as a metric from the point of view of a supervised ML model and the test statistic from the point of view of the hypothesis test. (In the case of the neural networks, this comes automatically by evaluating the loss at the end of training, since the metric is directly optimised). There is no sigmoid in the neural net model and a linear activation is used, while in the kernel approach it is integrated in the definition of the loss. - To understand the role of the different components of the NPLM methodology, we perform various tests. One of these is to replace the final metric, to be used as a test statistic, with more traditional ML ones. The sigmoid activation is used when necessary. These are variants of the NPLM method because the alternative is still data-driven but the test statistic is not the likelihood ratio.

--Referee 2 - unknown true distribution: Not knowing the true data-generating distributions is a common scenario in two-sample testing. Indeed, the goal is to test population-level hypotheses from data.

--Referee 2 - code: In this work we focus on comparing the NPLM methodology with different approaches to GoF. Codes for NPLM, in its kernel methods or neural network implementations, have been presented in earlier work and they are publicly available. We will update the repository with instruction on how to reproduce the results of this paper.

---

## Round 2 · List of Changes

--Referee 1 and 2: following the suggestions from the referees, we improved the overall presentation and writing throughout the paper. In the figures, we increased the font size, simplified the notation and improved the labels. Major changes involved: - abstract. - introduction: first paragraph. - caption figure 1. - central paragraph page 4. - paragraphs at end of page 5 and page 6. - conclusions. - final part of appendix A.2.

---

## Editorial Decision

published